# MVP: Multi-task Supervised Pre-training for Natural Language Generation

## Abstract

Pre-trained language models (PLMs) have achieved remarkable success in natural language generation (NLG) tasks. Up to now, most NLG-oriented PLMs are pre-trained in an unsupervised manner using the large-scale general corpus. In the meanwhile, an increasing number of models pre-trained with labeled data (*i.e.,* "*supervised pre-training*") showcase superior performance compared to unsupervised pre-trained models. Motivated by the success of supervised pre-training, we propose **M**ulti-task super**V**ised **P**re-training (**MVP**) for natural language generation. We collect a large-scale natural language generation corpus, MVPCorpus, from 77 datasets over 11 diverse NLG tasks. Then we unify these examples into a general text-to-text format to pre-train the text generation model MVP in a supervised manner. For each task, we further pre-train specific soft prompts to stimulate the model's capacity to perform a specific task. Extensive experiments have demonstrated the effectiveness and generalizability of our MVP model in a number of NLG tasks, which achieves state-of-the-art performance on 13 out of 17 datasets.

## 1 Introduction

Natural language generation (NLG, also known as text generation) is a crucial capacity for language intelligence, which aims to generate human-like texts on demand (Garbacea & Mei, 2020). Since the emergence of the pre-training and fine-tuning paradigm, pre-trained language models (PLMs) have dominated the mainstream approaches for NLG tasks (Lewis et al., 2020; Brown et al., 2020). With a large-scale general corpus, the majority of PLMs are pre-trained in an unsupervised (self-supervised) manner by leveraging intrinsic data correlations as supervision signals. However, unsupervised pre-training is likely to incorporate noise that affects the performance of downstream tasks (Feng et al., 2022), also leading to a slower rate of acquiring knowledge (Zhang et al., 2021).

In the meanwhile, more and more large-scale labeled datasets have become easily accessible (Deng et al., 2009; Liu et al., 2020). There is growing evidence that pre-training with labeled data can further improve the performance of PLMs, both in the fields of computer vision (He et al., 2016; Dosovitskiy et al., 2021) and natural language processing (Lin et al., 2020b; Su et al., 2022). These promising developments motivate us to consider pre-training text generation models with labeled data, which is called "*supervised pre-training*" (Feng et al., 2022). Existing work has shown that supervised pre-training can explicitly learn task-specific characteristics and alleviate the discrepancy between unsupervised pre-training and supervised fine-tuning (Sanh et al., 2022; Lin et al., 2020b).

Furthermore, most NLG systems are often trained in a supervised way, requiring supervision signals to learn the input-to-output transformation. For example, dialogue systems learn to generate appropriate responses based on historical utterances, and text summarization systems learn to extract essential information from long documents according to human-written summaries. Therefore, we suspect that supervised pre-training is more suited for NLG-oriented PLMs in essence since it can provide task-related instructions early in the *pre-training stage* instead of a later *fine-tuning stage*.

Inspired by the recent success of supervised pre-training, we propose **M**ulti-task super**V**ised **P**re-training (**MVP**) for natural language generation by leveraging a variety of labeled text generation datasets. Specially, we collect a large-scale labeled corpus, MVPCorpus, consisting of 77 datasets over 11 text generation tasks. Since recent research shows that an extensive scale of multi-task pre-training (Aribandi et al., 2022) is the key to generalizing to new tasks for large PLMs, we combine these labeled datasets for multi-task pre-training. Existing popular works, as shown in Table 1, mainly

Table 1: Representative PLMs for NLG and NLU tasks using (un)supervised pre-training. We present a more detailed comparison and discussion about supervised pre-training in Section 6.

| Settings | Supervised Pre-training | Unsupervised Pre-training |
|---|---|---|
| NLG | MVP (ours) | GPT-2, GPT-3, BART, T5, UniLM, MASS, PEGASUS |
| NLU | FLAN, T0, Muppet, ExT5 | BERT, RoBERTa, T5, UniLM, XLNet, ELECTRA |

focus on NLU tasks (Sanh et al., 2022; Aribandi et al., 2022) or use unsupervised pre-training (Lewis et al., 2020; Raffel et al., 2020), with no consideration of supervised pre-training on NLG tasks. To fill this gap, we explore supervised pre-training and multi-task learning for deriving both *effective* and *general* NLG models.

To develop our approach, we adopt a Transformer-based (Vaswani et al., 2017) sequence-to-sequence model as the pre-training backbone. In multi-task training, different tasks may "neutralize" the ability learned through other tasks (He & Choi, 2021). To mitigate this potential issue, we propose to learn task-specific prompts based on the MVP model, following the structure of prefix-tuning (Li & Liang, 2021). Task-specific pre-training enables prompts to "store" specialized knowledge for each corresponding task. Integrating MVP with task-specific prompts can further stimulate the model's capacity to perform some specific tasks.

To summarize, our main contributions center around the following research questions:

- *How to train an NLG-oriented PLM in a supervised pre-training way?* In order to prepare the supervised corpus, we collect a massive labeled MVPCorpus, consisting of 77 datasets over 11 NLG tasks across various domains and specific objectives. To the best of our knowledge, MVPCorpus is the largest collection of NLG datasets. Firstly, we formulate different NLG tasks as a general text-to-text form so that the supervised corpus can be used in a unified way for pre-training an NLG model. Our work presents a simple yet general approach for pre-training a more capable NLG model by leveraging various labeled NLG datasets.
- *Can supervised pre-trained NLG models be both effective and general?* Extensive experiments show that the supervised pre-trained MVP outperforms its unsupervised pre-trained counterpart BART in both full tuning ($+7.0\%$ on avarege) and parameter-efficient tuning ($+4.3\%$ on avarege) settings. Our MVP model achieves state-of-the-art performance on 13 out of 17 datasets. Furthermore, the experiments on unseen NLG and NLU tasks demonstrate that our supervised MVP model has a strong generalization ability for unseen tasks.

For reproducing and reusing our work, we release the collection MVPCorpus, the models (*e.g.,* MVP, task-specific prompts, and multi-task variants), and codes for pre-training and fine-tuning at the link: https://anonymous.4open.science/r/ICLR-2023-Paper3518/.

## 2 RELATED WORK

**Pre-trained Language Models.** Pre-trained language models have achieved exceptional success in a wide range of tasks, and the majority of them are pre-trained in an unsupervised manner (Brown et al., 2020; Devlin et al., 2019; Lewis et al., 2020; Raffel et al., 2020). For example, with large-scale plain texts as the unsupervised pre-training corpus, GPT-3 (Brown et al., 2020) employ language modeling as the pre-training task, *i.e.,* predicting the next token conditioned on previous tokens; BART (Lewis et al., 2020) learns to recover the original text from corrupted text which has been altered by arbitrary noise transformations. GPT-3 and BART use 570GB and 160GB of unlabeled text as the pre-training corpora, respectively. In the meanwhile, the computer vision community benefits a lot from the labeled dataset ImageNet (Deng et al., 2009). Influential models, such as ResNet (He et al., 2016) and ViT (Dosovitskiy et al., 2021), leverage ImageNet for pre-training. Inspired by the success of pre-training with labeled data, machine translation researchers explore supervised pre-training (McCann et al., 2017; Lin et al., 2020b). Lin et al. (2020b) attempt to pre-train a translation model with parallel data in multiple languages. Despite using much less pre-trained data, mRASP still achieves better performance than translation models pre-trained in an unsupervised manner (Lample & Conneau, 2019; Liu et al., 2020). In this paper, we propose to pre-train a universal NLG model in a supervised manner with collections of labeled datasets (23GB).

**Stage 1: Multi-task Supervised Pre-training**     **Stage 2: Task-specific Prompt Pre-training**

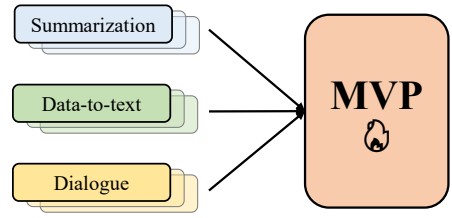 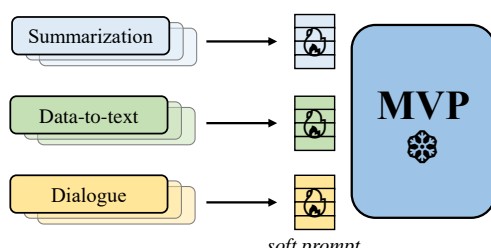

Figure 1: The overview of the pre-training process of our MVP model and task-specific prompts.

**Multi-task Learning.**   Our pre-training process is also related to multi-task learning (MTL), a method of mixing multiple tasks into a single training process (Collobert & Weston, 2008). A model trained with MTL can benefit from helpful knowledge of relevant tasks, resulting in improved performance (McCann et al., 2018; Subramanian et al., 2018). Recently, MT-DNN (Liu et al., 2019a) and Muppet (Aghajanyan et al., 2021) collect tens of datasets in the multi-task procedure and achieve better performance in downstream tasks. The *pre-finetuning* schema proposed in Muppet shares a similar idea with our study. Aribandi et al. (2022) further combine the denoising pre-training task of T5 (Raffel et al., 2020) and multi-task learning to pre-train a new model, ExT5. MTL has also contributed to sub-fields of text generation, such as open-ended dialogue system (Zhang et al., 2020), task-oriented dialogue system (Su et al., 2022), text style transfer (Bujnowski et al., 2020), and question answering (Khashabi et al., 2020). At the same time, researchers explore the transferability of models trained on multi-task datasets (Mishra et al., 2022). FLAN (Wei et al., 2022), T0 (Sanh et al., 2022), and ZeroPrompt (Xu et al., 2022) investigate the zero-shot generalization abilities of large PLMs trained on numerous task datasets with well-designed prompts. Ye et al. (2021) develop a benchmark CrossFit to study the few-shot learning ability of models. Compared with these works, we aim to explore multi-task learning to derive both *effective* and *general* NLG models.

**Prompt Learning.**   Prompt learning is a thriving method in the field of NLP. Prompt learning converts fine-tuning text into a format similar to pre-training to leverage implicit pre-training knowledge and alleviate the discrepancy between pre-training and fine-tuning (Liu et al., 2021b). GPT-2 (Radford et al., 2019) and T5 (Raffel et al., 2020) add human-written task prompts to the input text. For instance, T5 prepends "*Summarize:*" to the input document for summarization tasks; GPT-3 (Brown et al., 2020) further combines several demonstrations to input to learn task patterns, which is called *in-context learning*. Some researchers also design elaborate prompts or demonstrations for each task and dataset and investigate their effectiveness and robustness (Wei et al., 2022; Sanh et al., 2022; Xu et al., 2022; Mishra et al., 2022). To overcome the constraints of manually constructed prompts, researchers develop continuous (soft) prompts that can be optimized in the continuous space (Lester et al., 2021; Qin & Eisner, 2021). Prefix-tuning (Li & Liang, 2021) increases the number of parameters in prompts and employs prompting in each Transformer layer. Gu et al. (2022) propose PPT to pre-train continuous prompts using unlabeled data. SPoT (Vu et al., 2022) and UnifiedSKG (Xie et al., 2022) learn the prompts on related tasks and transfer them to new tasks.

## 3   THE MVP MODEL

This section introduces our **MVP** model: a **M**ulti-task super**V**ised **P**re-trained model for natural language generation. We first collect a large-scale NLG corpus, MVPCorpus, from 77 datasets over 11 diverse NLG tasks. After that, we pre-train our MVP model using a mixture of labeled data from MVPCorpus. We further learn the task-specific prompts to stimulate the MVP model to perform a certain task. The overview of our model is illustrated in Figure 1.

## 3.1 DATA COLLECTION

Formally, the natural language generation (NLG) task aims to generate a sequence of tokens $\mathcal{Y} = (y_1, y_2, \ldots, y_n)$ conditioned on input data $\mathcal{X}$ (*e.g.,* a piece of text or structured data) (Li et al., 2022). Typically, NLG tasks are categorized according to the data format of $\mathcal{X}$ and $\mathcal{Y}$. For example, text summarization condenses a long document into a brief text containing essential information; data-to-text generation produces descriptive text about structured input; and a dialogue system creates pertinent responses given multiple dialog utterances.

In this paper, we collect a large-scale labeled MVPCorpus consisting of 77 labeled datasets from 11 representative NLG tasks [1], including commonsense generation, data-to-text generation, open-ended dialogue system, paraphrase generation, question answering, question generation, story generation, task-oriented dialogue system, text simplification, text style transfer, and text summarization. These datasets come from various domains and are of different sizes. Some datasets are elaborately hand-crafted and thus relatively small in size, while others are created for large-scale weak supervision. Despite originating from various tasks, these diverse labeled datasets contain rich task-specific supervision signals for establishing global sequence-to-sequence mapping relations. The detailed descriptions of these tasks can be found in Appendix B.1.

Next, we transform all tasks into a unified text-to-text format and convert different input data $\mathcal{X}$ into a text format. For instance, we linearize structured data (*e.g.,* knowledge graph or table) by concatenating triples or key-value pairs using the special token "`[SEP]`" for data-to-text generation, and we utilize the special token "`[X_SEP]`" to separate answer and paragraph for question generation. The transformed input format for each task can be found in Appendix E.

We divide MVPCorpus into two parts, which are used for pre-training and fine-tuning (evaluation), respectively. For supervised pre-training, we utilize 50 datasets from 7 tasks, including data-to-text generation, open-ended dialogue system, question answering, question generation, story generation, task-oriented dialogue system, and text summarization. To enrich our pre-training corpus, we reverse the input and output of some tasks for obtaining new datasets (*e.g.,* story generation and summarization, question generation and question answering). We also eliminate pre-training examples overlapping with evaluation data to avoid data leakage (more details in Appendix B.2). Finally, we have a 25GB supervised pre-training corpus containing 32M examples. The statistics of datasets for pre-training are listed in Table 7.

For evaluation, we utilize the rest 27 datasets which are more commonly used in the literature. Among these datasets, 23 datasets are from the 7 tasks used in pre-training. We refer to them as *seen* tasks and use them to test the effectiveness of our model. The remaining 4 datasets are from the tasks of commonsense generation, paraphrase generation, simplification, and style transfer, respectively. We call them *unseen* tasks and use them to examine the generalization ability of our model.

## 3.2 MODEL ARCHITECTURE

We pre-train our MVP model and task-specific prompts in two stages. In the first stage, we pre-train the MVP backbone using a mixture of labeled datasets from seven tasks to learn general text-to-text relationships and transferable semantic information across tasks. To indicate each task, we apply human-written prompts to each task instance. For example, we write "*Summarize:*" as the prompt for summarization tasks. The manual prompts for each task are shown in Appendix E.

In the second stage, we freeze our MVP backbone and pre-train a set of task-specific soft prompts (*i.e.,* continuous vectors) to stimulate the model's capacity to perform some specific tasks. We learn them using a mixture of corresponding intra-task datasets (*i.e.,* datasets under the same task [2]). These soft prompts, which are not shared between tasks, encode the task-specific semantic knowledge to alleviate the blurring-out problem induced by multi-task learning (He & Choi, 2021).

Specifically, we employ the standard Transformer encoder-decoder (Vaswani et al., 2017) as our backbone. Compared to decoder-only architectures such as GPT-3 (Brown et al., 2020) and prefix LMs such as UniLM (Dong et al., 2019), the encoder-decoder architecture is more effective for

---

[1] We do not consider machine translation tasks but only focusing on English tasks in this work.

[2] For instance, we train summarization-specific prompts using summarization datasets (*e.g.,* Newsroom (Grusky et al., 2018), WikiHow (Koupaee & Wang, 2018), and MSNews (Liu et al., 2021a)).

text generation tasks (Raffel et al., 2020). As for task-specific soft prompts, we insert continuous vectors at each Transformer layer, following prefix-tuning (Li & Liang, 2021). Compared to prompt tuning (Lester et al., 2021), which only adds trainable embeddings to the input layer, the layer-wise prompting of prefix-tuning is more effective and stable (Liu et al., 2022), especially for NLG tasks.

### 3.3 TRAINING DETAILS

Our MVP model adopts a Transformer with 12 layers in both encoder and decoder (406M parameters), the same as the model size of BART$_{\text{LARGE}}$ (Lewis et al., 2020). The hidden size is $1,024$ and the inner hidden size of the feed-forward network is $4,096$. We employ the byte-pair-encoding (BPE) tokenizer, and the vocabulary size is $50,267$. We initialize the backbone with the BART parameters to provide a good starting point for NLG tasks following previous work (Dong et al., 2019; Zhang et al., 2020). We pre-train the model with a batch size of $8,192$ and adopt a temperature-scaled mixing strategy (Raffel et al., 2020) with a rate of $T = 2$ to mitigate the disparity in tasks and datasets.

We follow prefix-tuning (Li & Liang, 2021) to pre-train task-specific prompts by prepending trainable continuous vectors to the keys and values of the multi-head attention module at each layer. The prompt length is set to $100$, and we utilize the MLP reparameterization function with a hidden size of $800$ to improve the training robustness and performance (Li & Liang, 2021). Hence, every task prompts have approximately 62M parameters. Then, we freeze the MVP model and train seven groups of task-specific prompts, each of which corresponds to a different task. The batch size is set to $8,192$, and we leverage the mixing strategy with a rate of $T = 2$.

In the two stages, the maximum length of both input and output sequences is set to $1,024$ for supporting examples to contain more tokens. We optimize the model with a constant learning rate of $3 \times 10^{-5}$ using standard sequence-to-sequence cross-entropy loss. We apply the AdamW optimizer (Loshchilov & Hutter, 2019) with $\beta_1 = 0.9$, $\beta_2 = 0.98$, $\epsilon = 1 \times 10^{-6}$ to improve training stability (Liu et al., 2019b). The weight decay coefficient is $0.1$. For testing, we select the checkpoint with the highest validation performance. All the experiments are conducted on 32 NVIDIA Tesla V100 32GB GPUs. We implement our model using the library `Hugging Face` (Wolf et al., 2020).

In summary, we pre-train a 406M text generation model MVP and seven groups of 62M task-specific prompts. For each downstream task, users can either utilize the MVP backbone (406M) directly or further combine MVP with task-specific prompts (468M).

## 4 EXPERIMENT RESULTS

In this section, we mainly investigate the effectiveness of our proposed supervised pre-training for NLG. Specifically, we fine-tune our MVP model on new datasets for pre-trained (seen) generation tasks under *full tuning* and *parameter-efficient tuning* settings.

For the full tuning setting, we fine-tune the entire model (including the backbone MVP and prompts), while for the parameter-efficient tuning, we only fine-tune prompts but freeze the parameter weights of MVP. We optimize the model via the seq2seq loss with label smoothing (Szegedy et al., 2016) factor of 0.1 and the AdamW optimizer with default hyper-parameters. We sweep over the batch size in $\{16, 64, 256\}$ and the learning rate in $\{5 \times 10^{-6}, 1 \times 10^{-5}, 3 \times 10^{-5}\}$ to find the optimal hyper-parameters for each evaluation task. We utilize the checkpoint with the best validation performance for test set inference. During inference, we set the beam size to 5 and the no-repetitive ngram size to 3. For evaluation, we leverage the automatic generation metrics BLEU (Papineni et al., 2002), ROUGE (Lin, 2004), and METEOR (Banerjee & Lavie, 2005) to measure the quality of the generated text and employ Distinct (Li et al., 2016) to evaluate its diversity. Details regarding fine-tuning and evaluation can be found in Appendix C.

We conduct extensive experiments with in different settings. Under full tuning scenarios, we employ the 23 datasets from 7 seen tasks for evaluation. Section 4.1 and Appendix D analyze the performance of our methods on these datasets. To better compare with ExT5 (Aribandi et al., 2022), we conduct experiments on the GEM benchmark (Gehrmann et al., 2021) in Appendix D.2. Under parameter-efficient tuning settings, we utilize the same datasets as in Section 4.1 and the results can be found in Section 4.2. Furthermore, we evaluate our models without fine-tuning and compare them with T0 (Sanh et al., 2022) in Appendix D.3. These extensive results show that our MVP model consistently

Table 2: The main results on seven seen tasks under full tuning settings. The best and second-best results among all the methods are marked in **bold** and underlined, respectively. The SQuAD dataset here is used for the question generation task. The letters B, R, D, and ME denote BLEU, ROUGE, Distinct, and METEOR, respectively. "–" means the work does not compute the corresponding result. These setups and abbreviations are the same below. [a] (Ravaut et al., 2022) [b] (Ke et al., 2021) [c] (Bao et al., 2021) [d] (Xiao et al., 2020) [e] (Lewis et al., 2020) [f] (Liu et al., 2021a) [g] (Guan et al., 2021) [h] (Chen et al., 2022) [i] (He et al., 2022) [j] (Lin et al., 2020c)

| Methods | CNN/DailyMail | | | WebNLG | | | SQuAD (QG) | | | CoQA | |
|---|---|---|---|---|---|---|---|---|---|---|---|
| | R-1 | R-2 | R-L | B-4 | ME | R-L | B-4 | ME | R-L | F1 | EM |
| SOTA | **47.16**[a] | **22.55** | **43.87** | 66.14[b] | 47.25 | 76.10 | 25.97[c] | **27.33** | 53.43 | 84.50[d] | – |
| **MVP** | 44.44 | 21.61 | 40.99 | 67.76 | 47.74 | 77.04 | **26.21** | 27.20 | **53.46** | 86.51 | 77.62 |
| BART | 44.16[e] | 21.28 | 40.90 | 64.55[b] | 46.51 | 75.13 | 22.00[f] | 26.40 | 52.55 | 68.60[f] | – |
| Single | 44.36 | 21.54 | 40.88 | 67.74 | 46.89 | 76.94 | 26.09 | 27.15 | 53.29 | 86.20 | 77.26 |
| **MVP+S** | 44.30 | 21.53 | 40.83 | **68.21** | **47.77** | **77.09** | 26.13 | 27.18 | 53.44 | **86.75** | **77.97** |
| MVP+R | 44.14 | 21.45 | 40.72 | 67.61 | 47.65 | 76.70 | 25.71 | 27.03 | 53.09 | 85.95 | 77.22 |
| MVP+M | 43.97 | 21.16 | 40.46 | 67.45 | 47.57 | 76.81 | 25.46 | 26.79 | 52.95 | 86.28 | 77.26 |

| Methods | ROCStories | | | | PersonaChat | | | | MultiWOZ | | |
|---|---|---|---|---|---|---|---|---|---|---|---|
| | B-1 | B-2 | D-1 | D-4 | B-1 | B-2 | D-1 | D-2 | B-4 | Success | Inform |
| SOTA | 33.40[g] | 15.40 | – | 69.30 | 49.90[f] | 40.00 | 1.50[h] | 9.40 | **20.50**[i] | **85.30** | **94.40** |
| **MVP** | 33.42 | 15.54 | 2.92 | 75.06 | **50.07** | **40.54** | **1.54** | **9.86** | 20.04 | 79.20 | 87.20 |
| BART | 30.70[g] | 13.30 | – | 69.90 | 49.90[f] | 40.00 | 1.30 | 8.00 | 17.89[j] | 74.91 | 84.88 |
| Single | 32.67 | 15.29 | 2.72 | 72.97 | 49.96 | 40.53 | 1.27 | 7.63 | 19.73 | 75.60 | 83.70 |
| **MVP+S** | **33.92** | **15.60** | **3.44** | **80.58** | 47.64 | 39.84 | 1.40 | 8.10 | 19.91 | 77.60 | 84.60 |
| MVP+R | 32.93 | 15.32 | 2.88 | 73.83 | 48.45 | 40.09 | 1.30 | 7.95 | 19.02 | 73.30 | 81.80 |
| MVP+M | 33.30 | 15.51 | 2.71 | 74.24 | 46.26 | 39.30 | 1.36 | 8.07 | 19.93 | 72.70 | 79.70 |

outperforms various baselines in different scenarios, which demonstrates the effectiveness of our supervised pre-training method for NLG.

## 4.1 FULL TUNING PERFORMANCE

We design several model variants to verify the effectiveness of our two-stage pre-training method proposed in Section 3.2. For the first-stage model **MVP** that uses multi-task supervised pre-training, we compare it with two competitive backbones using different pre-training strategies:

- **BART$_{\text{LARGE}}$ (Lewis et al., 2020)**: BART is a widely-used PLM for natural language generation using unsupervised pre-training task, *i.e.,* denoising auto encoder.
- **Single-task pre-training (Single)**: We individually train a single model for each task using intra-task datasets under the same pre-training settings in multi-task training. For instance, we pre-train a summarization model using summarization datasets (*e.g.,* Newsroom, WikiHow, and MSNews). Therefore, we have seven single-task pre-trained models in total.

For the second-stage model that integrates pre-trained task-specific prompts (denoted by **MVP+S**), we compare it with two variants using different prompts:

- **Randomly initialized prompts (MVP+R)**: The layer-wise prompts for the MVP model are randomly initialized without pre-training.
- **Multi-Task pre-trained prompts (MVP+M)**: We only pre-train one group of prompts for all tasks, using the same mixed datasets as in the backbone pre-training.

Besides these variants, we further include the best-reported results from original papers in the literature for comparison (denoted as **SOTA**). From the results in Table 2, we can see that:

First, supervised pre-training models (*i.e.,* MVP and Single) achieve better performance than the unsupervised pre-trained model BART, yielding an average improvement of $7.0\%$ and $4.4\%$ (in ratio), respectively. This finding demonstrates the effectiveness of our supervised pre-training method. With

Table 3: The main results on seven seen tasks under parameter-efficient settings. We also include the results of BART and MVP under the full tuning setting (denoted as FT) for comparison.

| Methods | CNN/DailyMail | | | WebNLG | | | SQuAD (QG) | | | CoQA | |
|---|---|---|---|---|---|---|---|---|---|---|---|
| | R-1 | R-2 | R-L | B-4 | ME | R-L | B-4 | ME | R-L | F1 | EM |
| FT BART | 44.16 | 21.28 | 40.90 | 64.55 | 46.51 | 75.13 | 22.00 | 26.40 | 52.55 | 68.60 | – |
| FT MVP | 44.44 | 21.61 | 40.99 | 67.76 | 47.74 | 77.04 | 26.21 | 27.20 | 53.46 | 86.51 | 77.62 |
| **MVP+S** | **43.03** | 20.27 | **39.72** | **66.73** | **47.42** | **76.36** | **25.28** | **26.66** | 52.69 | **86.44** | **76.84** |
| BART+R | 42.47 | 19.82 | 39.15 | 65.54 | 46.86 | 75.24 | 24.27 | 26.07 | 52.03 | 82.22 | 71.92 |
| MVP+R | 42.84 | 20.21 | 39.61 | 66.12 | 47.12 | 75.83 | 25.05 | 26.34 | 52.57 | 85.51 | 75.56 |
| MVP+M | 42.99 | **20.36** | 39.70 | 66.40 | 47.16 | 75.89 | 25.24 | 26.49 | **52.88** | 85.90 | 76.34 |

| Methods | ROCStories | | | | PersonaChat | | | | MultiWOZ | | |
|---|---|---|---|---|---|---|---|---|---|---|---|
| | B-1 | B-2 | D-1 | D-4 | B-1 | B-2 | D-1 | D-2 | B-4 | Success | Inform |
| FT BART | 30.70 | 13.30 | – | 69.90 | 49.90 | 40.00 | 1.30 | 8.00 | 17.89 | 74.91 | 84.88 |
| FT MVP | 33.42 | 15.54 | 2.92 | 75.06 | 50.07 | 40.54 | 1.54 | 9.86 | 20.04 | 79.20 | 87.20 |
| **MVP+S** | **32.94** | 15.12 | **2.98** | **71.09** | **47.11** | **39.51** | **1.39** | **7.28** | **19.24** | **71.40** | **77.80** |
| BART+R | 32.14 | 14.71 | 2.85 | 68.94 | 46.23 | 38.98 | 1.30 | 6.82 | 17.94 | 62.20 | 69.20 |
| MVP+R | 32.28 | 14.85 | 2.97 | 70.29 | 46.70 | 39.23 | 1.31 | 6.98 | 18.86 | 64.40 | 71.40 |
| MVP+M | 32.62 | **15.28** | 2.95 | 69.58 | 46.78 | 39.40 | 1.33 | 7.13 | 19.13 | 67.20 | 72.90 |

labeled datasets, supervised pre-training enables the model to acquire more task-specific information, thus leading to improved results on downstream tasks. Regarding multi-task pre-training (MVP) and single-task (Single), our MVP model outperforms its single-task counterparts by 2.7%. This result indicates that the proposed multi-task learning approach can enhance single-task performance by learning transferable semantic information across tasks.

Second, task-specific prompt learning is effective to alleviate the "blurring-out" issue of multi-task learning. For tasks such as data-to-text generation and question answering, MVP with single-task prompt pre-training (MVP+S) consistently outperforms the other two variants (MVP+R, MVP+M). This verifies that task-specific prompts can acquire specialized knowledge of each task and stimulate the capacity of the MVP model to perform certain tasks.

Finally, our supervised pre-training approach achieves five new SOTA results on data-to-text generation, question generation, question answering, story generation, and open-ended dialogue tasks in Table 2. We also achieve SOTA performance in six out of eight datasets in Table 9, which shows the strong text generation capability of our MVP model. As for the remaining tasks, the SOTA models incorporate specific techniques tailored to the tasks, *e.g.,* the re-ranking framework (Ravaut et al., 2022) and various task-specific objectives (He et al., 2022), which yield better performance than our models. In contrast, the results of our models are very competitive, which is developed based on a general architecture and a unified learning objective.

## 4.2 PARAMETER-EFFICIENT TUNING PERFORMANCE

In the lightweight fine-tuning setting, we only tune the prompts while freezing the backbone MVP model. Besides our MVP+S model, we consider comparing the following methods:

- **Prefix-tuning** (Li & Liang, 2021): Prefix-tuning is a popular prompt-based lightweight tuning method for text generation. We employ BART$_{\text{LARGE}}$ as its backbone, denoted as **BART+R**.
- **Only tuning randomly initialized prompts (MVP+R)**: This variant only tunes the randomly initialized prompts of MVP+R, and it shares a similar idea with prefix-tuning.
- **Only tuning multi-task pre-trained prompts (MVP+M)**: This variant only tunes the multi-task pre-trained prompts of MVP+M. Such an idea has been used in SPoT (Vu et al., 2022).

From the experimental results in Table 3, we can see that: the good performance of the MVP model in lightweight settings further demonstrates the effectiveness of supervised pre-training. By comparing two randomly initialized prompting methods (BART+R and MVP+R), we can see that MVP+R achieves superior performance to BART+R (+2.0%) due to its multi-task supervised backbone. Furthermore, when initialized with pre-trained prompts, MVP+S and MVP+M achieve improved

Table 4: The main results of unseen NLG tasks. We use AESOP and SC & BLEU to denote the methods proposed by Sun et al. (2021) and Lai et al. (2021), respectively. Accuracy is calculated by a pre-trained TextCNN to evaluate the style strength, and HM denotes the harmonic mean of BLEU-4 and style accuracy. [a] (Sun et al., 2021) [b] (Lai et al., 2021)

| AESOP | Quora | | | | | SC & BLEU | GYAFC E&M | | | GYAFC F&R | | |
|---|---|---|---|---|---|---|---|---|---|---|---|---|
| | B-4 | R-1 | R-2 | R-L | ME | | B-4 | Accuracy | HM | B-4 | Accuracy | HM |
| +BART | 47.30[a] | 73.30 | 54.10 | 75.10 | 49.70 | +BART | 76.50[b] | 93.70 | 83.90 | 79.30 | 92.00 | 85.20 |
| +MVP | **49.86** | **74.93** | **56.55** | **76.56** | **52.27** | +MVP | **77.01** | **94.66** | **84.92** | **79.70** | **93.07** | **85.87** |

Table 5: The main results of NLU tasks on the GLUE benchmark. We evaluate the results on the official website `https://gluebenchmark.com/`. Matt. means the Matthews correlation coefficient. Acc. stands for the accuracy rate. P/S Corr. denote Pearson and Spearman correlation coefficients. m./mm. refer to the accuracy of the matched and mismatched domains. Avg. is a macro-average of scores defined in Wang et al. (2019).

| Methods | CoLA Matt. | SST-2 Acc. | MRPC F1/Acc. | STS-B P/S Corr. | QQP F1/Acc. | MNLI m./mm. | QNLI Acc. | RTE Acc. | Average |
|---|---|---|---|---|---|---|---|---|---|
| BART | **60.30** | 96.30 | 90.47 / 86.70 | 90.97 / 90.30 | 73.03 / 89.87 | **90.03 / 89.27** | 94.60 | 79.83 | 85.17 |
| MVP | 59.87 | **96.43** | **92.07 / 89.43** | **91.37 / 90.90** | **73.20 / 90.13** | 89.70 / 88.73 | **95.10** | **82.87** | **85.88** |

results over MVP+R, which is consistent with the findings of SPoT (Vu et al., 2022). When compared with MVP+M, MVP+S performs marginally better by $1.2\%$, indicating that task-specific prompts are useful to improve the model in specific generation tasks.

Surprisingly, our lightweight MVP+S can even outperform fully tuned BART on tasks such as question generation and question answering, showcasing the effectiveness of the proposed supervised pre-training approach. Another note is that lightweight prompting methods (Lester et al., 2021; Vu et al., 2022) that work on NLU tasks cannot achieve competitive performances when compared to full tuning methods on NLG tasks.

## 5 GENERALIZATION ABILITY

In this section, we test our MVP model on unseen NLG and NLU tasks to verify the generalizability.

**Generalization to Unseen NLG Tasks.** According to Deng et al. (2021), an NLG task can be assigned to one of the following three categories: compression (*e.g.,* summarization), transduction (*e.g.,* translation), or creation (*e.g.,* story generation). Since we do not include any transduction tasks during pre-training, we evaluate our MVP model using two unseen transduction NLG tasks: paraphrase generation and text style transfer. We select the SOTA methods for these two tasks, *i.e.,* AESOP (Sun et al., 2021) for paraphrase generation and SC & BLEU (Lai et al., 2021) for text style transfer, and replace their backbone BART with our MVP model for comparison. The experimental setup remains the same as described in Section 4, and details are reported in Appendix C. From the results in Table 4, we can see that our model outperforms BART by a ratio of $2.2\%$ and achieves two new SOTA results, which verifies the strong generalizability of our model. This finding shows that our MVP model is more capable than BART and can serve as a general yet effective backbone to solve more specific tasks by providing superior parameter initialization.

**Generalization to Unseen NLU Tasks.** Although MVP is designed especially for NLG tasks, we also evaluate its performance on unseen NLU tasks using the widely-used GLUE benchmark (Wang et al., 2019). We compare our model to BART$_{\text{LARGE}}$ using its original sequence classification method (Lewis et al., 2020). The detailed settings can be found in Appendix C. According to the results presented in Table 5, our MVP model outperforms BART on 9 of 12 metrics and has a superior overall performance of $0.71\%$. This result indicates the strong generalization ability of our MVP model and further demonstrates that our supervised pre-training not only learns generation ability but also improves the overall semantic representations.

Table 6: Comparison of our work with existing *supervised pre-training* methods. #NLG/#NLU denote the number of NLG and NLU tasks, respectively. PT denotes pre-training, FT denotes fine-tuning and SP denotes supervised pre-training.

| Methods | #NLG (PT) | #NLU (PT) | #NLG (FT) | #NLU (FT) | SP model | SP prompts | Open source |
|---|---|---|---|---|---|---|---|
| FLAN | 3 | 9 | 2 | 9 | ✓ | ✗ | ✗ |
| T0 | 2 | 6 | 0 | 4 | ✓ | ✗ | ✓ |
| Muppet | 1 | 3 | 1 | 3 | ✓ | ✗ | ✓ |
| ExT5 | 3 | 8 | 6 | 8 | ✓ | ✗ | ✗ |
| SPoT | 1 | 4 | 0 | 6 | ✗ | ✓ | ✗ |
| MVP (ours) | **7** | 0 | **11** | 3 | ✓ | ✓ | ✓ |

## 6  DISCUSSION

**Differences with Existing Methods.**   To the best of our knowledge, existing supervised pre-training works mainly focus on NLU tasks (Aghajanyan et al., 2021; Aribandi et al., 2022) or a small number of NLG tasks (Lin et al., 2020b; Su et al., 2022). Given the superior performance achieved by supervised pre-training approaches, it is important to explore supervised pre-training for deriving both *effective* and *general* NLG models. Our work makes a significant contribution in this direction, achieving SOTA performance with a single model on 13 of 17 datasets. Compared with its strong counterpart ExT5 (Aribandi et al., 2022), our MVP model outperforms it in 26 out of 27 metrics (detailed in Appendix D.2). In order to better understand the difference between our paper with previous supervised (multi-task) pre-training studies, we present a detailed comparison in Table 6. As we can see, our work conducts the study with the largest number of NLG tasks for both supervised pre-training and fine-tuning, incorporates task-specific prompts, and also releases all the important resources for reproducing or reusing our work.

**Applicability.**   To facilitate the application of our work, we have released the collection corpus, pre-trained models, task-specific prompts, and the generated texts. Our collected MVPCorpus is the largest NLG task collection. We can use all the data to pre-train a general model or select a subset to continue pre-training a domain- or task- specific model (Gururangan et al., 2020). Our MVPCorpus can also be considered as the evaluation benchmark for different NLG tasks. Furthermore, our MVP model can be used to achieve new state-of-the-art results in various NLG tasks. Users can either fine-tune the MVP model or integrate it with task-specific prompts to achieve better results based on sufficient labeled data. Even in data-scarce domains, our MVP model can be also directly employed to obtain good performance without fine-tuning. In addition, our MVP model can provide effective parameter initialization for improving existing methods, as described in Section 5. Finally, the pre-trained task-specific prompts and the generated texts can be further used to study the task similarity and their effect on the multi-task pre-training.

## 7  CONCLUSION

In this paper, we present **M**ulti-task super**V**ised **P**re-training (**MVP**) for natural language generation. Firstly, we collect a large-scale NLG corpus, MVPCorpus, from 77 datasets over 11 diverse NLG tasks. After converting various NLG tasks into a unified text-to-text format, we propose multi-task supervised pre-training to learn an *effective* and *general* model **MVP** with task-specific prompts for NLG tasks. Extensive experiments have demonstrated that: (1) supervised pre-training is beneficial for NLG tasks as a general solution. Our MVP model outperforms the unsupervised pre-trained model BART and even achieves SOTA performance on 13 out of 17 datasets; (2) supervised pre-trained models have strong generalization ability on unseen generation or even understanding tasks.

In future work, we will explore the multilingual version of our MVP model by covering more datasets in other languages. Such a model is expected to capture language-independent task characteristics and improve the generation tasks in the minority language. Besides, it is interesting to study how different tasks relate to each other in the unified semantic space of the MVP model, which can inspire methods that incorporate task relations as prior.

BROADER IMPACTS

In this paper, we pre-trained a language model MVP using labeled NLG datasets. According to the research (Bender et al., 2021; Bommasani et al., 2021), PLMs tend to "remember" what they have "seen" in pre-training corpus. This could result in the reproduction of undesirable biases from pre-training data on downstream tasks. Training data intervention could be a solution to alleviate this issue (Lu et al., 2020). It is also interesting to investigate whether supervised pre-training produces fewer biases than unsupervised pre-training in the future.

Environmental impact is another factor we should consider. We have attempted a more efficient pre-training strategy and released our PLM for future work. In contrast to large PLMs with tens of billions of parameters, such as T5 (Raffel et al., 2020) and GPT-3 (Brown et al., 2020), we pre-train only a small model with hundreds of millions of parameters. In addition, we utilize supervised pre-training data and initialize our model with pre-trained BART, both of which improve the convergence of our model. Ultimately, our model is pre-trained for about $20,000$ steps, whereas BART of the same size is pre-trained for $500,000$ steps.

REPRODUCIBILITY

For reproducing and reusing our work, we have released the collection MVPCorpus, the models (*e.g.,* MVP, task-specific prompts and multi-task variants), intermediate results (*e.g.,* the generated texts), and source codes for pre-training and fine-tuning at the link: `https://anonymous.4open.science/r/ICLR-2023-Paper3518/`. The detailed settings of experiments are listed in Appendix C. We hope that these open-source resources will facilitate future work on supervised pre-training and contribute to the advancement of NLG research.

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

## A  LIMITATIONS

Despite our efforts to collect as many generation tasks and datasets as possible, we only evaluate the generation quality and generalization ability of our models on a small number of tasks and datasets. The interpretability and robustness of our models require further analysis. Besides, there exists subjectivity when collecting intra-task datasets, albeit our attempts to employ widely-recognized categorizations from the literature. Due to limitation of computing power, we do not study the performance of our method at different model scales. The effectiveness of multi-task pre-training from scratch, similar to ExT5 (Aribandi et al., 2022), also merits an in-depth study. Regarding evaluation methods, we only consider basic automatic metrics such as BLEU (Papineni et al., 2002) and ROUGE (Lin, 2004). However, there is still a certain gap between these metrics and human judgments (Sai et al., 2022).

## B  TASKS AND DATASETS

### B.1  DESCRIPTION OF TASKS AND DATASETS

We provide the details of the tasks and datasets used in our paper for pre-training and fine-tuning in Tables 7 and 8. If the dataset for pre-training does not have a valid set, we divide 10% of the training set for validation.

We list the licenses for all datasets if them have. All datasets are publicly available. The majority of them can be directly downloaded from GitHub or Google Drive. ROCStories (Mostafazadeh et al., 2016) and CommonGen (Lin et al., 2020a) can be obtained after filling out a form. GYAFC (Rao & Tetreault, 2018) is accessible after requesting Yahoo and the authors of the dataset.

The tasks and datasets we use in this paper are as follows:

- **Data-to-text generation** aims to generate descriptive text about structured data, such as the knowledge graph and the table. We use the following datasets for pre-training:
    1. AGENDA (Koncel-Kedziorski et al., 2019);
    2. ENT-DESC (Cheng et al., 2020);
    3. GenWiki (Jin et al., 2020);
    4. LogicNLG (Chen et al., 2020a);
    5. TEKGEN (Agarwal et al., 2021);
    6. WEATHERGOV (Liang et al., 2009);
    7. WikiTableT (Chen et al., 2021).

  We utilize the following datasets for fine-tuning evaluation:
    1. WebNLG (Gardent et al., 2017), we utilize the version 2.1;
    2. WikiBio (Lebret et al., 2016).

- **Open-ended dialogue system**, also known as chatbots, is focused on daily communication. We use the following datasets for pre-training:
    1. Cleaned OpenSubtitles Dialogs (Cleaned OS Dialogs) (Welivita et al., 2021), which is a cleaned variant of OpenSubtitles Dialogs (Lison et al., 2018);
    2. CMU Document Grounded Conversations (CMUDog) (Zhou et al., 2018);
    3. Curiosity (Rodriguez et al., 2020);
    4. DREAM (Sun et al., 2019);
    5. Empathetic Dialogues (Rashkin et al., 2019);
    6. Movie Dialog (Dodge et al., 2016);
    7. MuTual (Stratos, 2019);
    8. OpenDialKG (Moon et al., 2019);
    9. Topical-Chat (Gopalakrishnan et al., 2019);
    10. Wizard of Wikipedia (Dinan et al., 2019).

  We utilize the following datasets for fine-tuning evaluation:

    1. DailyDialog (Li et al., 2017);

    2. DSTC7-AVSD (Alamri et al., 2018);

    3. PersonaChat (Zhang et al., 2018).

- **Paraphrase generation** involves rewriting a sentence with the same semantic meaning but a different syntactic or lexical form. We utilize the following datasets for fine-tuning evaluation:

    1. Quora (also known as QQP-Pos) (Kumar et al., 2020), which is a subset of Quora Question Pairs [3].

- **Question answering** requires the model to answer a question based on optional background information. Note that we conduct this task in a generative way in our paper. We use the following datasets for pre-training:

    1. HotpotQA (Yang et al., 2018);

    2. MS MARCO (Nguyen et al., 2016);

    3. MSQG (Liu et al., 2021a), since it is designed for QG, we reverse the question and answer to enrich QA examples;

    4. NarrativeQA (Kočiský et al., 2018);

    5. Natural Questions (Kwiatkowski et al., 2019);

    6. NewsQA (Trischler et al., 2017);

    7. QuAC (Choi et al., 2018);

    8. TriviaQA (Joshi et al., 2017);

    9. WebQuestions (Berant et al., 2013).

We utilize the following datasets for fine-tuning evaluation:

    1. CoQA (Reddy et al., 2019);

    2. SQuAD (Rajpurkar et al., 2016), we utilize the version 1.1.

- **Question generation** generates a coherent question given a passage and its corresponding answer. We use the following datasets for pre-training:

    1. HotpotQA (Yang et al., 2018);

    2. MS MARCO (Nguyen et al., 2016);

    3. MSQG (Liu et al., 2021a);

    4. NarrativeQA (Kočiský et al., 2018);

    5. NewsQA (Trischler et al., 2017);

    6. QuAC (Choi et al., 2018);

Most of them are QA tasks, and we invert the question and answer to enrich QG examples.

We utilize the following datasets for fine-tuning evaluation:

    1. CoQA (Reddy et al., 2019);

    2. SQuAD (Rajpurkar et al., 2016), we utilize the version 1.1.

- **Story generation** creates a long and informative text with a short title. We use the following datasets for pre-training:

    1. ChangeMyView (Hua & Wang, 2020);

    2. English Gigaword (Rush et al., 2015);

    3. Hippocorpus (Sap et al., 2020);

    4. WikiPlots (wik);

    5. WritingPrompts (Fan et al., 2018), we split the original training set for pre-training and corresponding validation.

Considering English Gigaword is a large summarization dataset, we use the summary as the title to generate the passage in turn to enrich the examples of story generation.

We utilize the following datasets for fine-tuning evaluation:

    1. ROCStories (Mostafazadeh et al., 2016);

---

[3] https://www.kaggle.com/c/quora-question-pairs

2. WritingPrompts (Fan et al., 2018), we use the sets created by Guan et al. (2021) (who split the original valid and test sets for training, validation, and testing) to fine-tune our model for a fair comparison.

- **Task-oriented dialogue system** meets real-life needs of users, such as restaurant reservations and airplane bookings. We use the datasets for pre-training, following Su et al. (2022):

    1. CamRest676 (Wen et al., 2017);
    2. Frames (El Asri et al., 2017);
    3. KVRET (Eric et al., 2017);
    4. MetaLWOZ (Lee et al., 2019);
    5. MSR-E2E (Li et al., 2018);
    6. MultiWOZ (Budzianowski et al., 2018);
    7. Schema-Guided (Rastogi et al., 2020a);
    8. TaskMaster (Byrne et al., 2019);
    9. WOZ (Mrkšić et al., 2017).

    We utilize the following datasets for fine-tuning evaluation:

    1. MultiWOZ (Budzianowski et al., 2018), we utilize the version 2.0;

- **Text style transfer** modifies the style (*e.g.,* sentiment and formality) of given texts while retaining their style-independent content. We utilize the following datasets for fine-tuning evaluation:

    1. GYAFC (Rao & Tetreault, 2018), which has two sub-domains "Entertainment and Music" (E&M) and "Family and Relationships" (F&R).

- **Text summarization** condenses a long document into a brief text while retaining the essential details. We use the following datasets for pre-training:

    1. English Gigaword (Graff et al., 2003), we use the variant provided by Rush et al. (2015);
    2. MediaSum (Zhu et al., 2021);
    3. MSNews (Liu et al., 2021a);
    4. Newsroom (Grusky et al., 2018);
    5. WikiHow (Koupaee & Wang, 2018).

    We utilize the following datasets for fine-tuning evaluation:

    1. CNN/DailyMail (Hermann et al., 2015), we use the variant provided by See et al. (2017);
    2. SAMSum (Gliwa et al., 2019);
    3. XSum (Narayan et al., 2018).

To better compare with ExT5 (Aribandi et al., 2022), we utilize the language generation benchmark GEM (Gehrmann et al., 2021) for fine-tuning evaluation. GEM includes five tasks:

- **Commonsense generation**:

    1. CommonGen (CG) (Lin et al., 2020a).

- **Data-to-text generation**:

    1. DART (Nan et al., 2021);
    2. E2E NLG cleaned (Novikova et al., 2017);
    3. ToTTo (Su et al., 2021);
    4. WebNLG (Gardent et al., 2017).

- **Dialogue system**:

    1. Schema-Guided Dialog (SGD) (Rastogi et al., 2020b).

- **Text simplification**:

    1. WikiAuto + Turk/ASSET (WiA-T/A) (Jiang et al., 2020; Xu et al., 2016; Alva-Manchego et al., 2020).

- **Text summarization**:

    1. Wiki-Lingua (WLE) (Ladhak et al., 2020).

To test the generalization ability of our model, we also utilize the natural language standing benchmark GLUE (Wang et al., 2019), which is composed of three tasks:

- **Natural language inference**:
    1. MNLI (Williams et al., 2018);
    2. QNLI (Rajpurkar et al., 2016; Wang et al., 2019);
    3. RTE (Dagan et al., 2006; Haim et al., 2006; Giampiccolo et al., 2007; Bentivogli et al., 2009).
- **Paraphrase detection**:
    1. MRPC (Dolan & Brockett, 2005);
    2. QQP [3];
    3. STS-B (Cer et al., 2017).
- **Text classification**:
    1. CoLA (Warstadt et al., 2019);
    2. SST-2 (Socher et al., 2013).

## B.2    DATA LEAKAGE

Since our model is pre-trained on a large number of labeled datasets, it may have "seen" examples from fine-tuning test sets during pre-training, which leads to an unfair comparison with other methods. Hence, we eliminate the pre-training examples that share $n$-gram overlap with either of the test datasets. Following Brown et al. (2020), $n$ is the $5^{\text{th}}$ percentile example length in words, and the maximum value of $n$ is set to 13. Finally, we have removed $17,848$ examples from the pre-training datasets. The number of "cleaned" examples for each dataset can be found in Table 7.

Table 7: The statistics and licenses of datasets for pre-training our MVP model. The #Train, #Valid, and #Test denote the number of examples in the train, valid, and test sets, respectively. Cleaned #Train represents the number of training examples after filtering. Input and Output are the average number of words (split by space) in the input and output sequences, respectively. These setups and abbreviations are the same below.

| Dataset | #Train | Cleaned #Train | #Valid | #Test | Input | Output | License |
|---|---|---|---|---|---|---|---|
| AGENDA | 38,720 | 38,720 | 1,000 | 1,000 | 52.1 | 141.2 | N/A |
| ENT-DESC | 88,652 | 88,652 | 11,081 | 11,081 | 279.9 | 31.0 | N/A |
| GenWiki | 681,436 | 681,436 | 75,716 | 1,000 | 21.4 | 29.5 | MIT |
| LogicNLG | 28,450 | 28,450 | 4,260 | 4,305 | 178.4 | 14.2 | MIT |
| TEKGEN | 6,310,061 | 6,307,995 | 788,746 | 796,982 | 17.0 | 21.2 | CC BY-SA 2.0 |
| WEATHERGOV | 25,000 | 25,000 | 1,000 | 3,528 | 148.7 | 30.6 | N/A |
| WikiTableT | 1,453,794 | 1,452,778 | 4,533 | 4,351 | 81.0 | 99.7 | MIT |
| Cleaned OS Dialogs | 13,355,487 | 13,355,368 | 1,483,944 | - | 75.5 | 16.7 | N/A |
| CMUDoG | 82,818 | 82,818 | 5,555 | 14,510 | 433.0 | 12.2 | N/A |
| Curiosity | 64,930 | 64,551 | 8,539 | 8,495 | 144.4 | 20.2 | CC BY-NC 4.0 |
| DREAM | 14,264 | 14,242 | 4,709 | 4,766 | 75.6 | 13.6 | N/A |
| Empathetic Dialogues | 64,636 | 64,636 | 9,308 | 8,426 | 52.7 | 12.9 | CC BY-NC 4.0 |
| Movie Dialog | 762,751 | 762,711 | 8,216 | 8,066 | 126.9 | 44.0 | N/A |
| MuTual | 33,691 | 33,691 | 4,090 | 3,248 | 53.6 | 14.5 | N/A |
| OpenDialKG | 69,680 | 69,680 | 7,743 | - | 54.2 | 12.4 | CC BY-NC 4.0 |
| Topical-Chat | 179,750 | 179,750 | 22,295 | 22,452 | 223.3 | 20.0 | CDLA-Sharing-1.0 |
| Wizard of Wikipedia | 148,357 | 147,702 | 15,767 | 15,564 | 297.0 | 16.7 | MIT |
| HotpotQA | 90,447 | 87,815 | 7,405 | - | 187.9 | 2.2 | CC BY-SA 4.0 |
| MS MARCO | 681,445 | 681,226 | 77,580 | - | 68.7 | 13.3 | N/A |
| MSQG | 198,058 | 198,029 | 11,008 | - | 48.1 | 3.7 | CC BY-SA 4.0 |
| NarrativeQA | 65,494 | 65,494 | 6,922 | 21,114 | 584.1 | 4.2 | Apache 2.0 |
| Natural Questions | 96,676 | 96,676 | 10,693 | 6,490 | 9.0 | 2.1 | CC BY-SA 3.0 |
| NewsQA | 97,850 | 97,700 | 5,486 | 5,396 | 726.8 | 5.0 | MIT |
| QuAC | 83,568 | 83,485 | 31,906 | - | 487.9 | 12.5 | CC BY-SA 4.0 |
| TriviaQA | 78,785 | 78,785 | 8,837 | 11,313 | 14.0 | 2.0 | Apache 2.0 |
| WebQuestions | 8,933 | 8,933 | 4,863 | 4,863 | 6.7 | 2.4 | CC BY 4.0 |
| HotpotQA | 90,440 | 87,808 | 6,972 | - | 79.6 | 19.8 | CC BY-SA 4.0 |
| MS MARCO | 681,445 | 681,226 | 77,580 | - | 75.9 | 6.0 | N/A |
| MSQG | 198,058 | 198,029 | 11,008 | 11,022 | 45.9 | 6.0 | CC BY-SA 4.0 |
| NarrativeQA | 65,494 | 65,494 | 6,922 | 21,114 | 579.7 | 8.6 | Apache 2.0 |
| NewsQA | 97,850 | 97,700 | 5,486 | 5,396 | 724.2 | 7.6 | MIT |
| QuAC | 69,109 | 69,026 | 26,301 | - | 496.7 | 6.5 | CC BY-SA 4.0 |
| ChangeMyView | 42,462 | 42,459 | 6,480 | 7,562 | 17.9 | 104.1 | MIT |
| English Gigaword | 3,803,957 | 3,802,620 | 189,651 | 1,951 | 8.8 | 33.3 | MIT |
| Hippocorpus | 6,168 | 6,168 | 686 | - | 34.1 | 262.6 | CDLA-Permissive 2.0 |
| WikiPlots | 101,642 | 101,641 | 11,294 | - | 3.4 | 338.5 | N/A |
| WritingPrompts | 272,600 | 272,518 | 15,620 | 15,138 | 28.4 | 630.8 | MIT |
| CamRest676 | 4,872 | 4,872 | 616 | - | 55.3 | 9.4 | N/A |
| Frames | 26,631 | 26,631 | 2,106 | - | 116.1 | 13.0 | MIT |
| KVRET | 14,136 | 14,136 | 1,616 | - | 30.5 | 9.3 | N/A |
| MetaLWOZ | 176,073 | 176,073 | 17,912 | - | 45.6 | 8.0 | N/A |
| MSR-E2E | 103,362 | 103,362 | 5,235 | - | 51.3 | 12.8 | Microsoft |
| Schema-Guided | 494,946 | 494,933 | 73,089 | - | 120.8 | 12.5 | CC BY-SA 4.0 |
| TaskMaster | 249,664 | 249,662 | 20,680 | - | 95.6 | 12.0 | CC BY 4.0 |
| WOZ | 6,364 | 6,359 | 1,260 | - | 47.0 | 10.6 | N/A |
| English Gigaword | 3,803,957 | 3,802,620 | 189,651 | 1,951 | 33.3 | 8.8 | MIT |
| MediaSum | 443,596 | 442,021 | 10,000 | 10,000 | 1641.0 | 14.4 | N/A |
| MSNews | 136,082 | 135,937 | 7,496 | 7,562 | 309.9 | 9.8 | CC BY-SA 4.0 |
| Newsroom | 995,041 | 989,351 | 108,837 | 108,862 | 642.4 | 26.7 | N/A |
| WikiHow | 157,252 | 157,247 | 5,599 | 5,577 | 502.6 | 45.6 | CC BY-NC-SA |

Table 8: The statistics and licenses of datasets for evaluating our MVP model. The license of the MNLI dataset is composed of OANC, CC BY-SA 3.0, and CC BY 3.0. The license of the CoQA dataset is composed of CC BY-SA 4.0, MSR-LA, and Apache 2.0. The license of the WiA-A/T datasets is composed of CC BY-NC 3.0, CC BY-NC 4.0, and GNU General Public License v3.0.

| Task | Dataset | #Train | #Valid | #Test | Input | Output | License |
|---|---|---|---|---|---|---|---|
| **Commonsen generation** | CommonGen | 67,389 | 993 | – | 5.5 | 11.6 | MIT |
| **Data-to-text generation** | DART | 62,659 | 2,768 | – | 27.5 | 21.5 | MIT |
| | E2E | 33,525 | 4,299 | – | 9.5 | 20.6 | CC BY-SA 4.0 |
| | ToTTo | 120,761 | 7,700 | – | 37.8 | 18.0 | CC BY-SA 3.0 |
| | WebNLG | 34,338 | 4,313 | 4,222 | 18.0 | 19.9 | CC BY-NA-SA 4.0 |
| | WebNLG (GEM) | 35,426 | 1,667 | – | 17.7 | 22.7 | CC BY-NA-SA 4.0 |
| | WikiBio | 582,659 | 72,831 | 72,831 | 81.6 | 26.1 | CC BY-SA 3.0 |
| **Open-ended dialogue** | DailyDialog | 76,052 | 7,069 | 6,740 | 72.5 | 13.9 | CC BY-NC-SA 4.0 |
| | DSTC7-AVSD | 76,590 | 17,870 | 1,710 | 148.2 | 11.5 | MIT |
| | PersonaChat | 122,499 | 14,602 | 14,056 | 132.1 | 11.9 | MIT |
| | SGD | 164,982 | 10,000 | – | 134.7 | 11.3 | CC BY-SA 4.0 |
| **Natural language inference** | MNLI-m | 392,702 | 9,815 | 9,796 | 29.8 | – | Mixed |
| | MNLI-mm | | 9,832 | 9,847 | | | |
| | QNLI | 104,743 | 5,463 | 5,463 | 36.6 | – | CC BY-SA 4.0 |
| | RTE | 2,490 | 277 | 3,000 | 51.0 | – | N/A |
| **Paraphrase generation** | Quora | 137,185 | 3,000 | 3,000 | 10.9 | 10.8 | N/A |
| **Paraphrase detection** | MRPC | 3,668 | 408 | 1,725 | 43.8 | – | N/A |
| | QQP | 363,846 | 40,430 | 390,965 | 22.3 | – | N/A |
| | STS-B | 5,749 | 1,500 | 1,379 | 20.3 | – | N/A |
| **Question answering** | CoQA | 107,286 | 31,621 | – | 349.4 | 2.6 | Mixed |
| | SQuAD | 75,722 | 10,570 | 11,877 | 156.2 | 3.6 | CC BY-SA 4.0 |
| **Question generation** | CoQA | 107,286 | 31,621 | – | 346.6 | 5.5 | Mixed |
| | SQuAD | 75,722 | 10,570 | 11,877 | 148.3 | 11.6 | CC BY-SA 4.0 |
| **Story generation** | ROCStories | 176,688 | 9,816 | 4,909 | 9.0 | 40.7 | N/A |
| | WritingPrompts | 53,516 | 4,000 | 2,000 | 25.5 | 150.4 | MIT |
| **Task-oriented dialogue** | MultiWOZ | 170,220 | 22,074 | 22,116 | 128.3 | 11.3 | MIT |
| **Text classification** | CoLA | 8,551 | 1,043 | 1,063 | 7.7 | – | N/A |
| | SST-2 | 67,349 | 872 | 1,821 | 9.8 | – | N/A |
| **Text simplification** | WiA-A | 483,801 | 20,000 | 359 | 26.2 | 21.5 | Mixed |
| | WiA-T | | | 359 | | | |
| **Text style transfer** | GYAFC-E&M | 52,595 | 11,508 | 1,416 | 9.9 | 10.6 | N/A |
| | GYAFC-F&R | 51,967 | 11,152 | 1,332 | 10.7 | 11.3 | |
| **Text summarization** | CNN/DailyMail | 287,227 | 13,368 | 11,490 | 679.8 | 48.3 | MIT |
| | SAMSum | 14,732 | 818 | 819 | 103.4 | 20.3 | CC BY-NC-ND 4.0 |
| | WLE | 99,020 | 28,614 | – | 367.6 | 33.4 | CC0 1.0 |
| | XSum | 204,045 | 11,332 | 11,334 | 373.7 | 21.1 | MIT |

## C  FINE-TUNING AND EVALUATION DETAILS

In this section, we introduce the details for fine-tuning and evaluating each downstream task.

For the experiments in Section 4 (Tables 2 and 3), and Appendix D.1 (Table 9), the fine-tuning details are introduced in Section 4, and the evaluation details are presented as follows:

- For data-to-text generation tasks, we use BLEU(-4), ROUGE-L, and METEOR for evaluation. We use the script provided by Chen et al. (2020b) [4];
- For open-ended dialogue system tasks, we use BLEU-1, BLEU-2, Distinct-1, and Distinct-2 for evaluation. For DSTC7-AVSD we also utilize CIDEr (Vedantam et al., 2015). We employ NLTK 3.5 with smoothing function 7 to compute BLEU for PersonaChat and DailyDialog, and utilize the script [5] to evaluate DSTC7-AVSD;
- For question answering tasks, we use Exact Match (EM) and Macro-averaged F1 score (F1) for evaluation. We use the provided script for CoQA [6] and SQuAD [7].
- For question generation tasks, we use BLEU-4, ROUGE-L, and METEOR for evaluation. We use the script provided by Dong et al. (2019) [8];
- For story generation, we employ nucleus sampling with $p = 0.9$ and temperature of $0.7$ following Guan et al. (2021). We use corpus BLEU-1, BLEU-2, Distinct-1, and Distinct-4 for evaluation. We use NLTK 3.5 to calculate corpus BLEU following Guan et al. (2021);
- For task-oriented dialogue system tasks, we use BLEU(-4), inform (rate), success (rate), and combined score for evaluation. Inform and success are two specially designed accuracy metrics for task-oriented dialogue system, and the combined score is defined as (Inform + Success) $\times 0.5 +$ BLEU (Budzianowski et al., 2018). We use the script provided by Su et al. (2022) [9];
- For text summarization tasks, we use ROUGE-1, ROUGE-2, and ROUGE-L for evaluation. We use the toolkit `files2rouge` [10].

For the experiments in Section 5 (Tables 4 and 5), the fine-tuning and evaluation details are as follows:

- For paraphrase generation tasks, we employ the fine-tuning and evaluation scripts provided by AESOP (Sun et al., 2021) [11]. The evaluation metrics are BLEU-4, ROUGE-1, ROUGE-2, ROUGE-L, and METEOR.
- For text style transfer tasks, we employ the fine-tuning and evaluation scripts provided by SC & BLEU (Lai et al., 2021) [12]. We conduct the informal-to-formal transfer and train the model on the data from both the E&M and F&R domains following Lai et al. (2021). The evaluation metrics are BLEU-4, accuracy, and HM. Accuracy is calculated by a pre-trained TextCNN to evaluate the style strength, and HM denotes the harmonic mean of BLEU-4 and style accuracy (Lai et al., 2021).
- For GLUE tasks, we utilize the fine-tuning code provided by Hugging Face [13]. The hyper-parameters are consistent with original BART (Lewis et al., 2020) [14]. The evaluation is computed by the official website [15].

---

[4] https://github.com/wenhuchen/Data-to-text-Evaluation-Metric
[5] https://github.com/lemuria-wchen/DialogVED/blob/main/src/utils/evaluate.py
[6] https://github.com/PaddlePaddle/ERNIE/blob/repro/ernie-gen/eval/tasks/coqa/eval.py
[7] https://github.com/allenai/bi-att-flow/blob/master/squad/evaluate-v1.1.py
[8] https://github.com/microsoft/unilm/blob/master/unilm-v1/src/qg/eval.py
[9] https://github.com/awslabs/pptod/blob/main/E2E_TOD/eval.py
[10] https://github.com/pltrdy/files2rouge
[11] https://github.com/PlusLabNLP/AESOP
[12] https://github.com/laihuiyuan/pre-trained-formality-transfer
[13] https://github.com/huggingface/transformers/tree/main/examples/pytorch/text-classification
[14] https://github.com/facebookresearch/fairseq/blob/main/examples/bart/README.glue.md
[15] https://gluebenchmark.com/

Table 9: The results on six seen tasks under full tuning settings. [a] (Nguyen et al., 2021) [b] (Tang et al., 2022) [c] (Gu et al., 2021) [d] (Lewis et al., 2020) [e] (Guan et al., 2021) [f] (Chen et al., 2022) [g] (Chen et al., 2020b) [h] (Raffel et al., 2020) [i] (Xu et al., 2021)

| Methods | XSum | | | SAMSum | | | CoQA QG | | |
|---|---|---|---|---|---|---|---|---|---|
| | R-1 | R-2 | R-L | R-1 | R-2 | R-L | B-4 | ME | R-L |
| SOTA | **49.57**[a] | **25.08** | **41.81** | 53.89[b] | 28.85 | 49.29 | 15.78[c] | 40.15 | 50.98 |
| BART | 45.14[d] | 22.27 | 37.25 | 51.74[b] | 26.46 | 48.72 | 12.34[c] | 35.78 | 46.88 |
| MVP | 45.67 | 22.53 | 37.41 | 53.91 | 29.28 | 49.40 | 22.96 | **47.14** | 54.84 |
| MVP+S | 45.59 | 22.54 | 37.39 | **53.97** | **29.44** | **49.69** | **23.04** | 47.01 | **55.14** |

| Methods | WritingPrompts | | | | DailyDialog | | | | WikiBio |
|---|---|---|---|---|---|---|---|---|---|
| | B-1 | B-2 | D-1 | D-4 | B-1 | B-2 | D-1 | D-2 | B-4 |
| SOTA | 22.40[e] | 8.40 | – | 31.30 | 46.10[f] | 40.70 | 4.10 | 22.20 | 45.10[g] |
| BART | 22.40[e] | 8.40 | – | 31.30 | 44.30[f] | 39.20 | 3.90 | 21.10 | – |
| MVP | **31.81** | **12.80** | 2.58 | 69.45 | **52.34** | **43.93** | 6.39 | 35.65 | **48.42** |
| MVP+S | 29.18 | 11.11 | **3.71** | **78.02** | 51.04 | 42.87 | **6.70** | **36.84** | 48.19 |

| Methods | DSTC7-AVSD | | | | | | | SQuAD | |
|---|---|---|---|---|---|---|---|---|---|
| | B-1 | B-2 | B-3 | B-4 | ME | R-L | CIDEr | F1 | EM |
| SOTA | 83.20[f] | 70.50 | 59.80 | 50.60 | 31.40 | 63.80 | 1.391 | **96.22**[h] | **91.26** |
| BART | 82.40[f] | 69.10 | 58.20 | 48.70 | 31.30 | 63.50 | 1.382 | 91.56[i] | 84.23 |
| MVP | **83.92** | 70.83 | 60.06 | 50.64 | **31.95** | **64.87** | **1.429** | 93.04 | 86.44 |
| MVP+S | 83.82 | **70.95** | **60.23** | **50.87** | 31.49 | 64.29 | 1.409 | 93.21 | 86.78 |

For the experiments of the GEM benchmark in Appendix D.2 (Table 10), the fine-tuning settings are the same as those described in Section 4. We use BLEU-4, ROUGE-2, and METEOR for evaluation. We use the GEM evaluation scripts [16].

# D    ADDITIONAL RESULTS

In this section, we provide additional results of our MVP model and other baselines.

## D.1    RESULTS OF COMMON DATASETS

We also conduct experiments on eight common datasets under full tuning settings. Due to space limits in Section 4, these results are shown in Table 9. We can see that these results share a similar trend to those in Section 4, and we achieve SOTA performances in 6 of 8 datasets.

## D.2    RESULTS ON THE GEM BENCHMARK

To better compare with ExT5 (Aribandi et al., 2022), we conduct experiments on the GEM benchmark (Gehrmann et al., 2021). For "unseen" commonsense generation and text simplification tasks, we utilize prompts of data-to-text generation and summarization, respectively. The results are presented in Table 10, and our MVP models outperform ExT5 in 26 out of 27 metrics.

## D.3    RESULTS WITHOUT FINE-TUNING

Considering our MVP model has already been pre-trained on several tasks, we conduct experiments on these "seen" tasks without fine-tuning our model. To some degree, this setting can be viewed as

---

[16]https://github.com/GEM-benchmark/GEM-metrics

Table 10: The results on the GEM benchmark under full tuning settings. We utilize the large version of T5.1.1 and ExT5, and all the results of them are from Aribandi et al. (2022).

| Methods | DART | | | E2E | | | ToTTo | | |
|---|---|---|---|---|---|---|---|---|---|
| | B-4 | R-2 | ME | B-4 | R-2 | ME | B-4 | R-2 | ME |
| T5.1.1 | 34.31 | 45.22 | 36.30 | **42.57** | 46.60 | 38.20 | 39.79 | 49.90 | 36.80 |
| ExT5 | 36.62 | 48.14 | 37.60 | 42.25 | 46.70 | 38.10 | 40.14 | 50.33 | 36.90 |
| MVP | **39.13** | **48.92** | **38.53** | 37.38 | **47.96** | **39.39** | 50.58 | 55.24 | 41.27 |
| MVP+S | 38.83 | 48.49 | 38.41 | 37.32 | 47.40 | 38.90 | **50.69** | **55.52** | **41.29** |

| Methods | WebNLG | | | CommonGen | | | SGD | | |
|---|---|---|---|---|---|---|---|---|---|
| | B-4 | R-2 | ME | B-4 | R-2 | ME | B-4 | R-2 | ME |
| T5.1.1 | 31.67 | 43.31 | 34.40 | 8.38 | 17.01 | 20.20 | 33.15 | 36.17 | 32.40 |
| ExT5 | 35.03 | 48.17 | 36.50 | 9.68 | 19.04 | 21.40 | 34.74 | 37.77 | 33.00 |
| MVP | **47.03** | 59.00 | **42.34** | 32.59 | 37.71 | 33.00 | **45.63** | **48.29** | **48.48** |
| MVP+S | **47.03** | **59.03** | 42.28 | **34.10** | **37.87** | **33.11** | 45.24 | 48.25 | 38.47 |

| Methods | WiA-A | | | WiA-T | | | WLE | | |
|---|---|---|---|---|---|---|---|---|---|
| | B-4 | R-2 | ME | B-4 | R-2 | ME | B-4 | R-2 | ME |
| T5.1.1 | 29.30 | 38.37 | 30.10 | 42.12 | 50.52 | 36.2 | 15.55 | 20.47 | 19.60 |
| ExT5 | 29.23 | 37.98 | 30.00 | 41.39 | 50.38 | 35.8 | 16.64 | 21.16 | 20.40 |
| MVP | **71.55** | **70.88** | **48.19** | **91.73** | 83.46 | **57.34** | **18.80** | **22.84** | 21.95 |
| MVP+S | 70.37 | 70.65 | 47.70 | 91.12 | **83.59** | 56.95 | 18.52 | 22.57 | **22.02** |

Table 11: The results on seven seen tasks without fine-tuning. Given that T0 has been pre-trained on the CNN/DailyMail dataset, we exclude their results to provide a fair comparison (denoted as "–").

| Methods | CNN/DailyMail | | | WebNLG | | | SQuAD (QG) | | | CoQA | |
|---|---|---|---|---|---|---|---|---|---|---|---|
| | R-1 | R-2 | R-L | B-4 | ME | R-L | B-4 | ME | R-L | F1 | EM |
| FT BART | 44.16 | 21.28 | 40.90 | 64.55 | 46.51 | 75.13 | 22.00 | 26.40 | 52.55 | 68.60 | – |
| FT MVP | 44.44 | 21.61 | 40.99 | 67.76 | 47.74 | 77.04 | 26.21 | 27.20 | 53.46 | 86.51 | 77.62 |
| T0 | – | – | – | 1.40 | 10.20 | 18.43 | 3.06 | 12.43 | 14.91 | 13.30 | 6.60 |
| MVP | **29.50** | **11.29** | **25.92** | 34.42 | 31.33 | 52.33 | 2.90 | 13.94 | 15.48 | 29.40 | 18.20 |
| MVP+S | 25.60 | 9.51 | 22.67 | **39.43** | **34.32** | **55.34** | **2.96** | **15.23** | **18.23** | **52.40** | **37.30** |

| Methods | ROCStories | | | | PersonaChat | | | | MultiWOZ | | |
|---|---|---|---|---|---|---|---|---|---|---|---|
| | B-1 | B-2 | D-1 | D-4 | B-1 | B-2 | D-1 | D-2 | B-4 | Success | Inform |
| FT BART | 30.70 | 13.30 | – | 69.90 | 49.90 | 40.00 | 1.30 | 8.00 | 17.89 | 74.91 | 84.88 |
| FT MVP | 33.42 | 15.54 | 2.92 | 75.06 | 50.07 | 40.54 | 1.54 | 9.86 | 20.04 | 79.20 | 87.20 |
| T0 | 8.69 | 3.02 | 4.37 | 35.49 | 23.20 | 23.57 | 2.56 | 12.06 | 0.02 | **2.50** | 22.10 |
| MVP | 1.01 | 0.31 | **7.18** | **86.26** | 35.54 | 32.71 | **2.87** | **16.38** | **3.08** | **2.50** | **22.20** |
| MVP+S | **10.52** | **3.54** | 2.13 | 69.55 | **37.04** | **33.38** | 2.66 | 14.84 | 0.38 | **2.50** | 22.10 |

zero-shot learning. Nonetheless, it does not conform to the definition of *true zero-shot* settings (Perez et al., 2021). To avoid controversy, we refer to this as *without fine-tuning*.

We include T0-3B (Sanh et al., 2022) as our baseline. The results are listed in Table 11. Our MVP model outperforms T0 in all metrics with a large margin. However, all tasks demonstrate that methods without fine-tuning perform significantly worse than those with full tuning settings. This suggests that zero-shot strategies that are effective for NLU tasks may not produce satisfactory results for NLG tasks. Even though our model has acquired task knowledge, it struggles to perform well in a new domain without being fine-tuned. Thus, we focus mainly on full tuning settings in this paper.

# E    QUALITATIVE EXAMPLES

In this section, we showcase the linearized inputs, human-written task prompts, and corresponding outputs of a single dataset for tasks in Section 4. We provide the results of BART, MVP, and MVP+S under full tuning settings. To minimize human intervention, we select the first and second instances of the test set.

Table 12: The first instance from the CNN/Daily Mail dataset. Human-written task prompts are labeled in *italic*. The setting is the same below.

**Input**

*Summarize:* Marseille, France (CNN)The French prosecutor leading an investigation into the crash of Germanwings Flight 9525 insisted Wednesday that he was not aware of any video footage from on board the plane. Marseille prosecutor Brice Robin told CNN that "so far no videos were used in the crash investigation." He added, "A person who has such a video needs to immediately give it to the investigators." Robin's comments follow claims by two magazines, German daily Bild and French Paris Match, of a cell phone video showing the harrowing final seconds from on board Germanwings Flight 9525 as it crashed into the French Alps. All 150 on board were killed. Paris Match and Bild reported that the video was recovered from a phone at the wreckage site. The two publications described the supposed video, but did not post it on their websites. The publications said that they watched the video, which was found by a source close to the investigation. "One can hear cries of 'My God' in several languages," Paris Match reported. "Metallic banging can also be heard more than three times, perhaps of the pilot trying to open the cockpit door with a heavy object. Towards the end, after a heavy shake, stronger than the others, the screaming intensifies. Then nothing." "It is a very disturbing scene," said Julian Reichelt, editor-in-chief of Bild online. An official with France's accident investigation agency, the BEA, said the agency is not aware of any such video. Lt. Col. Jean-Marc Menichini, a French Gendarmerie spokesman in charge of communications on rescue efforts around the Germanwings crash site, told CNN that the reports were "completely wrong" and "unwarranted." Cell phones have been collected at the site, he said, but that they "hadn't been exploited yet." Menichini said he believed the cell phones would need to be sent to the Criminal Research Institute in Rosny sous-Bois, near Paris, in order to be analyzed by specialized technicians working hand-in-hand with investigators. But none of the cell phones found so far have been sent to the institute, Menichini said. Asked whether staff involved in the search could have leaked a memory card to the media, Menichini answered with a categorical "no." Reichelt told "Erin Burnett: Outfront" that he had watched the video and stood by the report, saying Bild and Paris Match are "very confident" that the clip is real. He noted that investigators only revealed they'd recovered cell phones from the crash site after Bild and Paris Match published their reports. "That is something we did not know before. ... Overall we can say many things of the investigation weren't revealed by the investigation at the beginning," he said. What was mental state of Germanwings co-pilot? German airline Lufthansa confirmed Tuesday that co-pilot Andreas Lubitz had battled depression years before he took the controls of Germanwings Flight 9525, which he's accused of deliberately crashing last week in the French Alps. Lubitz told his Lufthansa flight training school in 2009 that he had a "previous episode of severe depression," the airline said Tuesday. Email correspondence between Lubitz and the school discovered in an internal investigation, Lufthansa said, included medical documents he submitted in connection with resuming his flight training. The announcement indicates that Lufthansa, the parent company of Germanwings, knew of Lubitz's battle with depression, allowed him to continue training and ultimately put him in the cockpit. Lufthansa, whose CEO Carsten Spohr previously said Lubitz was 100% fit to fly, described its statement Tuesday as a "swift and seamless clarification" and said it was sharing the information and documents – including training and medical records – with public prosecutors. Spohr traveled to the crash site Wednesday, where recovery teams have been working for the past week to recover human remains and plane debris scattered across a steep mountainside. He saw the crisis center set up in Seyne-les-Alpes, laid a wreath in the village of Le Vernet, closer to the crash site, where grieving families have left flowers at a makeshift stone memorial. Menichini told CNN late Tuesday that no visible human remains were left at the site but recovery teams would keep searching. French President Francois Hollande, speaking Tuesday, said that it should be possible to identify all the victims using DNA analysis by the end of the week, sooner than authorities had previously suggested. In the meantime, the recovery of the victims' personal belongings will start Wednesday, Menichini said. Among those personal belongings could be more cell phones belonging to the 144 passengers and six crew on board. Check out the latest from our correspondents. The details about Lubitz's correspondence with the flight school during his training were among several developments as investigators continued to delve into what caused the crash and Lubitz's possible motive for downing the jet. A Lufthansa spokesperson told CNN on Tuesday that Lubitz had a valid medical certificate, had passed all his examinations and "held all the licenses required." Earlier, a spokesman for the prosecutor's office in Dusseldorf, Christoph Kumpa, said medical records reveal Lubitz suffered from suicidal tendencies at some point before his aviation career and underwent psychotherapy before he got his pilot's license. Kumpa emphasized there's no evidence suggesting Lubitz was suicidal or acting aggressively before the crash. Investigators are looking into whether Lubitz feared his medical condition would cause him to lose his pilot's license, a European government official briefed on the investigation told CNN on Tuesday. While flying was "a big part of his life," the source said, it's only one theory being considered. Another source, a law enforcement official briefed on the investigation, also told CNN that authorities believe the primary motive for Lubitz to bring down the plane was that he feared he would not be allowed to fly because of his medical problems. Lubitz's girlfriend told investigators he had seen an eye doctor and a neuropsychologist, both of whom deemed him unfit to work recently and concluded he had psychological issues, the European government official said. But no matter what details emerge about his previous mental health struggles, there's more to the story, said Brian Russell, a forensic psychologist. "Psychology can explain why somebody would turn rage inward on themselves about the fact that maybe they weren't going to keep doing their job and they're upset about that and so they're suicidal," he said. "But there is no mental illness that explains why somebody then feels entitled to also take that rage and turn it outward on 149 other people who had nothing to do with the person's problems." Germanwings crash compensation: What we know. Who was the captain of Germanwings Flight 9525? CNN's Margot Haddad reported from Marseille and Pamela Brown from Dusseldorf, while Laura Smith-Spark wrote from London. CNN's Frederik Pleitgen, Pamela Boykoff, Antonia Mortensen, Sandrine Amiel and Anna-Maja Rappard contributed to this report.

**Gold**
Match are "very confident" the video clip is real, an editor says. Andreas Lubitz had informed his Lufthansa training school of an episode of severe depression, airline says.

**BART**
Prosecutor: "A person who has such a video needs to immediately give it to the investigators" French Gendarmerie spokesman: Cell phones have been collected at the site, but haven't been exploited.

**MVP**
Prosecutor: "So far no videos were used in the crash investigation" An official with France's accident investigation agency says the agency is not aware of any video.

**MVP+S**
Prosecutor: "So far no videos were used in the crash investigation" Two publications claim to have found a cell phone video of the crash. "It is a very disturbing scene," says editor-in-chief of Bild online.

Table 13: The second instance from the CNN/Daily Mail dataset.

**Input**

*Summarize:* The Palestinian Authority officially became the 123rd member of the International Criminal Court on Wednesday, a step that gives the court jurisdiction over alleged crimes in Palestinian territories. The formal accession was marked with a ceremony at The Hague, in the Netherlands, where the court is based. The Palestinians signed the ICC's founding Rome Statute in January, when they also accepted its jurisdiction over alleged crimes committed "in the occupied Palestinian territory, including East Jerusalem, since June 13, 2014." Later that month, the ICC opened a preliminary examination into the situation in Palestinian territories, paving the way for possible war crimes investigations against Israelis. As members of the court, Palestinians may be subject to counter-charges as well. Israel and the United States, neither of which is an ICC member, opposed the Palestinians' efforts to join the body. But Palestinian Foreign Minister Riad al-Malki, speaking at Wednesday's ceremony, said it was a move toward greater justice. "As Palestine formally becomes a State Party to the Rome Statute today, the world is also a step closer to ending a long era of impunity and injustice," he said, according to an ICC news release. "Indeed, today brings us closer to our shared goals of justice and peace." Judge Kuniko Ozaki, a vice president of the ICC, said acceding to the treaty was just the first step for the Palestinians. "As the Rome Statute today enters into force for the State of Palestine, Palestine acquires all the rights as well as responsibilities that come with being a State Party to the Statute. These are substantive commitments, which cannot be taken lightly," she said. Rights group Human Rights Watch welcomed the development. "Governments seeking to penalize Palestine for joining the ICC should immediately end their pressure, and countries that support universal acceptance of the court's treaty should speak out to welcome its membership," said Balkees Jarrah, international justice counsel for the group. "What's objectionable is the attempts to undermine international justice, not Palestine's decision to join a treaty to which over 100 countries around the world are members." In January, when the preliminary ICC examination was opened, Israeli Prime Minister Benjamin Netanyahu described it as an outrage, saying the court was overstepping its boundaries. The United States also said it "strongly" disagreed with the court's decision. "As we have said repeatedly, we do not believe that Palestine is a state and therefore we do not believe that it is eligible to join the ICC," the State Department said in a statement. It urged the warring sides to resolve their differences through direct negotiations. "We will continue to oppose actions against Israel at the ICC as counterproductive to the cause of peace," it said. But the ICC begs to differ with the definition of a state for its purposes and refers to the territories as "Palestine." While a preliminary examination is not a formal investigation, it allows the court to review evidence and determine whether to investigate suspects on both sides. Prosecutor Fatou Bensouda said her office would "conduct its analysis in full independence and impartiality." The war between Israel and Hamas militants in Gaza last summer left more than 2,000 people dead. The inquiry will include alleged war crimes committed since June. The International Criminal Court was set up in 2002 to prosecute genocide, crimes against humanity and war crimes. CNN's Vasco Cotovio, Kareem Khadder and Faith Karimi contributed to this report.

**Gold**

Membership gives the ICC jurisdiction over alleged crimes committed in Palestinian territories since last June. Israel and the United States opposed the move, which could open the door to war crimes investigations against Israelis.

**BART**

Palestinian Authority becomes 123rd member of the International Criminal Court. The move gives the court jurisdiction over alleged crimes in Palestinian territories. Israel and the United States opposed the Palestinians' efforts to join the body.

**MVP**

"Today brings us closer to our shared goals of justice and peace," foreign minister says. The Palestinians signed the ICC's founding Rome Statute in January. The move gives the court jurisdiction over alleged crimes in Palestinian territories.

**MVP+S**

"Today brings us closer to our shared goals of justice and peace," foreign minister says. The United States says it "strongly" disagrees with the decision. The Palestinian Authority is the 123rd member of the International Criminal Court.

Table 14: The first instance from the WebNLG dataset, which has two gold target sentences.

| **Input** |
| --- |
| *Describe the following data:* Abilene,_Texas — cityServed — Abilene_Regional_Airport |

| **Gold** |
| --- |
| Abilene, Texas is served by the Abilene regional airport. |
| Abilene Regional Airport serves the city of Abilene in Texas. |

| **BART** |
| --- |
| Abilene Regional Airport serves the city of Abilene in Texas. |

| **MVP** |
| --- |
| Abilene Regional Airport serves the city of Abilene, Texas. |

| **MVP+S** |
| --- |
| Abilene Regional Airport serves the city of Abilene, Texas. |

Table 15: The second instance from the WebNLG dataset, which has three gold target sentences.

| **Input** |
| --- |
| *Describe the following data:* "Madrid, Paracuellos de Jarama, San Sebastián de los Reyes and Alcobendas" — location — Adolfo_Suárez_Madrid–Barajas_Airport |

| **Gold** |
| --- |
| Adolfo Suárez Madrid–Barajas Airport can be found in Madrid, Paracuellos de Jarama, San Sebastián de los Reyes and Alcobendas. |
| Adolfo Suarez Madrid-Barajas airport is located at Madrid, Paracuellos de Jarama, San Sebastián de los Reyes and Alcobendas. |
| Adolfo Suarez Madrid-Barajas Airport is located in Madrid, Paracuellos de Jarama, San Sebastian de los Reyes and Alcobendas. |

| **BART** |
| --- |
| Adolfo Suárez Madrid–Barajas Airport can be found in Madrid, Paracuellos de Jarama, San Sebastián de los Reyes and Alcobendas. |

| **MVP** |
| --- |
| Adolfo Suárez Madrid–Barajas Airport can be found in Madrid, Paracuellos de Jarama, San Sebastián de los Reyes and Alcobendas. |

| **MVP+S** |
| --- |
| Adolfo Suárez Madrid–Barajas Airport is located in Madrid, Paracuellos de Jarama, San Sebastián de los Reyes and Alcobendas. |

Table 16: The first instance from the SQuAD dataset.

**Input**

*Generate the question based on the answer:* Saint Bernadette Soubirous [SEP] Architecturally , the school has a Catholic character . Atop the Main Building ' s gold dome is a golden statue of the Virgin Mary . Immediately in front of the Main Building and facing it , is a copper statue of Christ with arms upraised with the legend " Venite Ad Me Omnes " . Next to the Main Building is the Basilica of the Sacred Heart . Immediately behind the basilica is the Grotto , a Marian place of prayer and reflection . It is a replica of the grotto at Lourdes , France where the Virgin Mary reputedly appeared to Saint Bernadette Soubirous in 1858 . At the end of the main drive ( and in a direct line that connects through 3 statues and the Gold Dome ) , is a simple , modern stone statue of Mary .

**Gold**

To whom did the Virgin Mary allegedly appear in 1858 in Lourdes France ?

**BART**

Who is believed to have appeared to the Virgin Mary at Lourdes ?

**MVP**

Who did the Virgin Mary appear to in Lourdes ?

**MVP+S**

The Grotto is a replica of the grotto at Lourdes , France where the Virgin Mary appeared to whom ?

Table 17: The second instance from the SQuAD dataset.

**Input**

*Generate the question based on the answer:* a copper statue of Christ [SEP] Architecturally , the school has a Catholic character . Atop the Main Building ' s gold dome is a golden statue of the Virgin Mary . Immediately in front of the Main Building and facing it , is a copper statue of Christ with arms upraised with the legend " Venite Ad Me Omnes " . Next to the Main Building is the Basilica of the Sacred Heart . Immediately behind the basilica is the Grotto , a Marian place of prayer and reflection . It is a replica of the grotto at Lourdes , France where the Virgin Mary reputedly appeared to Saint Bernadette Soubirous in 1858 . At the end of the main drive ( and in a direct line that connects through 3 statues and the Gold Dome ) , is a simple , modern stone statue of Mary .

**Gold**

What is in front of the Notre Dame Main Building ?

**BART**

What is in front of the Main Building and facing it ?

**MVP**

What is immediately in front of the Main Building ?

**MVP+S**

What is immediately in front of the Main Building ?

Table 18: The first instance from the CoQA dataset.

**Input**

*Answer the following question:* what color was cotton ? [X_SEP] once upon a time , in a barn near a farm house , there lived a little white kitten named cotton . cotton lived high up in a nice warm place above the barn where all of the farmer ' s horses slept . but cotton wasn ' t alone in her little home above the barn , oh no . she shared her hay bed with her mommy and 5 other sisters . all of her sisters were cute and fluffy , like cotton . but she was the only white one in the bunch . the rest of her sisters were all orange with beautiful white tiger stripes like cotton ' s mommy . being different made cotton quite sad . she often wished she looked like the rest of her family . so one day , when cotton found a can of the old farmer ' s orange paint , she used it to paint herself like them . when her mommy and sisters found her they started laughing . " what are you doing , cotton ? ! " " i only wanted to be more like you " . cotton ' s mommy rubbed her face on cotton ' s and said " oh cotton , but your fur is so pretty and special , like you . we would never want you to be any other way " . and with that , cotton ' s mommy picked her up and dropped her into a big bucket of water . when cotton came out she was herself again . her sisters licked her face until cotton ' s fur was all all dry . " don ' t ever do that again , cotton ! " they all cried . " next time you might mess up that pretty white fur of yours and we wouldn ' t want that ! " then cotton thought , " i change my mind . i like being special " .

**Gold**
white

**BART**
white

**MVP**
white

**MVP+S**
white

Table 19: The second instance from the CoQA dataset.

**Input**

*Answer the following question:* what color was cotton ? [SEP] white [X_SEP] where did she live ? [X_SEP] once upon a time , in a barn near a farm house , there lived a little white kitten named cotton . cotton lived high up in a nice warm place above the barn where all of the farmer ' s horses slept . but cotton wasn ' t alone in her little home above the barn , oh no . she shared her hay bed with her mommy and 5 other sisters . all of her sisters were cute and fluffy , like cotton . but she was the only white one in the bunch . the rest of her sisters were all orange with beautiful white tiger stripes like cotton ' s mommy . being different made cotton quite sad . she often wished she looked like the rest of her family . so one day , when cotton found a can of the old farmer ' s orange paint , she used it to paint herself like them . when her mommy and sisters found her they started laughing . " what are you doing , cotton ? ! " " i only wanted to be more like you " . cotton ' s mommy rubbed her face on cotton ' s and said " oh cotton , but your fur is so pretty and special , like you . we would never want you to be any other way " . and with that , cotton ' s mommy picked her up and dropped her into a big bucket of water . when cotton came out she was herself again . her sisters licked her face until cotton ' s fur was all all dry . " don ' t ever do that again , cotton ! " they all cried . " next time you might mess up that pretty white fur of yours and we wouldn ' t want that ! " then cotton thought , " i change my mind . i like being special " .

**Gold**
in a barn

**BART**
in a barn

**MVP**
in a barn

**MVP+S**
in a barn

Table 20: The first instance from the ROCStories dataset.

**Input**
*Given the story title:* male was out jogging one morning .

**Gold**
the weather was crisp and cool . male felt good and energetic . he decided to keep jogging longer than normal . male went several more miles out of his way .

**BART**
he saw a man struggling to walk across a field . male stopped to help him . he saw a guy struggling to walking across a fields . the man thanked male profusely !

**MVP**
he ran into a man who was holding a knife . male was scared , so he ran away . the man threatened male . male ran back into his house .

**MVP+S**
he noticed a sound coming from a tree . male went to investigate . he found a small baby bird stuck in the trunk . male took the bird to the vet and they released it .

Table 21: The second instance from the ROCStories dataset.

**Input**
*Given the story title:* ken was driving around in the snow .

**Gold**
he needed to get home from work . he was driving slowly to avoid accidents . unfortunately the roads were too slick and ken lost control . his tires lost traction and he hit a tree .

**BART**
he hit a patch of black ice . ken slid and hurt his knee . he went to the doctor for it . krampus was sent home with a concussion .

**MVP**
he was in his car and it was very cold . ken started to back up . the car started to make a loud noise . kena had to back out and call for help .

**MVP+S**
he was having trouble starting his car . he had to push it a couple of blocks . his car would n't start . ken had to call a tow truck for help .

Table 22: The first instance from the PersonaChat dataset.

**Input**
*Given the dialog:* i love to meet new people . [SEP] i have a turtle named timothy . [SEP] my favorite sport is ultimate frisbee . [SEP] my parents are living in bora bora . [SEP] autumn is my favorite season . [X_SEP] hello , how are you doing tonight ?

**Gold**
i am well an loving this interaction how are you ?

**BART**
i am doing well , how are you ?

**MVP**
i am doing well . how are you ?

**MVP+S**
i am doing well . how are you ?

Table 23: The second instance from the PersonaChat dataset.

**Input**
*Given the dialog:* i just bought a brand new house . [SEP] i like to dance at the club . [SEP] i run a dog obedience school . [SEP] i have a big sweet tooth . [SEP] i like taking and posting selkies . [X_SEP] hello , how are you doing tonight ? [SEP] i am well an loving this interaction how are you ?

**Gold**
i am great . i just got back from the club .

**BART**
i am doing well , thanks for asking . what do you do for fun ?

**MVP**
i am doing well . i just got home from dancing at the club .

**MVP+S**
i am doing well . i just bought a new house .

Table 24: The first instance from the MultiWOZ dataset. The task-oriented dialogue is composed of dialogue state tracking, dialogue action learning and system response generation.

**Input**

*Given the task dialog:* Belief state [X_SEP] i would like a taxi from saint john 's college to pizza hut fen ditton .

*Given the task dialog:* Dialogue action [X_SEP] [db_nores] [X_SEP] i would like a taxi from saint john 's college to pizza hut fen ditton .

*Given the task dialog:* System response [X_SEP] [db_nores] [X_SEP] i would like a taxi from saint john 's college to pizza hut fen ditton .

**Gold**

[taxi] destination pizza hut fen ditton departure saint john 's college

[taxi] [request] leave arrive

what time do you want to leave and what time do you want to arrive by ?

**BART**

[taxi] destination pizza express fen ditton departure saint johns college

[taxi] [request] leave

what time would you like to leave or arrive ?

**MVP**

[taxi] destination pizza hut fen ditton departure saint john 's college

[taxi] [request] leave

sure ! when would you like to arrive ?

**MVP+S**

[taxi] destination pizza hut fen ditton departure saint john 's college

[taxi] [request] leave arrive

what time would you like to leave ?

Table 25: The second instance from the MultiWOZ dataset.

**Input**

*Given the task dialog:* Belief state [X_SEP] i would like a taxi from saint john 's college to pizza hut fen ditton . [SEP] what time do you want to leave and what time do you want to arrive by ? [SEP] i want to leave after 17:15 .

*Given the task dialog:* Dialogue action [X_SEP] [db_nores] [X_SEP] i would like a taxi from saint john 's college to pizza hut fen ditton . [SEP] what time do you want to leave and what time do you want to arrive by ? [SEP] i want to leave after 17:15 .

*Given the task dialog:* System response [X_SEP] [db_nores] [X_SEP] i would like a taxi from saint john 's college to pizza hut fen ditton . [SEP] what time do you want to leave and what time do you want to arrive by ? [SEP] i want to leave after 17:15 .

**Gold**

[taxi] destination pizza hut fen ditton departure saint john 's college leave 17:15

[taxi] [inform] car phone

booking completed ! your taxi will be [value_car] contact number is [value_phone]

**BART**

[taxi] destination pizza hut fen ditton departure saint john 's college leave 17:15

[taxi] [inform] car phone

booking completed ! booked car type : [value_car] contact number : [value_phone]

**MVP**

[taxi] destination pizza hut fen ditton departure saint john 's college leave 17:15

[taxi] [inform] car phone

booking completed ! booked car type : [value_car] contact number : [value_phone]

**MVP+S**

[taxi] destination pizza hut fen ditton departure saint john 's college leave 17:15

[taxi] [inform] car phone

booking completed ! booked car type : [value_car] contact number : [value_phone]

