# OpenReview forum: "MVP: Multi-task Supervised Pre-training for Natural Language Generation"
_ICLR.cc/2023/Conference — Submitted to ICLR 2023_

### Official Review · Reviewer_4XxP · 2022-10-25

**Confidence:** 3
**Correctness:** 3
**Technical Novelty And Significance:** 2
**Empirical Novelty And Significance:** 3
**Recommendation:** 5

**Clarity, Quality, Novelty And Reproducibility:**

- On table 10, it seems there are a large improvement against ExT5 on the GEM benchmark. Since the ExT5 paper doesn't release their model checkpoints, and the evaluation script of GEM benchmark can make a big difference on the results, it would be good to add BART finetune baseline on Table 10 too -- so that we can exclude the effect of evaluation difference.

**Strength And Weaknesses:**

Strengths:
- A collection of data mixtures over 77 NLG datasets, which are publicly released.
- A exploration effort on studying multi-task pretraining on the NLG area.
- Code are released and could be easy to reproduce the results.

Weaknesses:
- The idea of multi-task pretraining is not novel, as it mostly follows Aribandi et al. (2022).
- Results are mixed. On Table 2, it seems the multi-task pretraining has minor gains over BART finetuning or Single; while on Table 9, the improvements are indeed significant.
- The proposed variants of pretraining stages (e.g., MVP+S, MVP+R, MVP+M) don't seem effective as MVP (the results are on-par on most datasets from Table 2 & 3)

**Summary Of The Paper:**

This paper focuses on multi-task pretraining via supervised data in NLG area. They collected 77 NLG datasets and applied different variants of multi-task pretraining on top. The pretrained models are evaluated on both seen and unseen tasks, comparing to the prior STOAs and BART finetune baselines.

**Summary Of The Review:**

Overall, the paper makes a significant contribution on the collection of a large amount of NLG datasets and study the multi-task pretraining behaviors for NLG. Their code are released and the results can be reproduced. The idea presented in this paper is only marginally novel and mostly follow the prior work. The results of their proposed method also seems mixed in different datasets.

---

> ### Author Response · Authors · 2022-11-14
> **Response to Reviewer 4XxP**
>
>
>
> Thank you for your meaningful reviews!
>
> - For the mixed results in Tables 2, 9, and 10. Your concerns are right. We copied the results of BART and ExT5 from existing papers for faithfulness because they may have specific optimal hyper-parameters that we cannot reproduce. The datasets in Table 2 are used more frequently than the ones in Tables 9 and 10. Some of the evaluation details cannot be found or have a large gap. Following your advice, we will include the results of BART that we reproduced.
>
> - For the proposed variants (e.g., MVP+S, MVP+R, and MVP+M). MVP+R and MVP+M are used for ablation analysis. In Tables 2 and 3, MVP+S consistently outperforms MVP+R and MVP+M, which demonstrates the effectiveness of our single-task pre-trained prompts. Compared with MVP, MVP+S achieves superior performance in tasks such as data-to-text generation, question answering, and story generation under full tuning settings. Moreover, MVP+S is an effective and efficient method under parameter-efficient settings.

---

> > ### Author Response · Authors · 2022-12-06
> > **Response to Reviewer 4XxP**
> >
> > We have added the human evaluation results. We are still very willing to discuss this with you in order to fully address your concerns. Please feel free to contact us with any additional suggestions or questions. We will keep doing our best to provide you with answers.

---

### Official Review · Reviewer_AMUT · 2022-10-25

**Confidence:** 4
**Correctness:** 1
**Technical Novelty And Significance:** 1
**Empirical Novelty And Significance:** 1
**Recommendation:** 1

**Clarity, Quality, Novelty And Reproducibility:**

- Clarity: It should be emphasized from the beginning (including abstract) that this is a study of continued multi-task supervised pre-training, and not a replacement for unsupervised pre-training.
- Reproducibility: Missing details about how different tasks are mixed in supervised pre-training.
- Novelty: continued pre-training with related tasks is not new, and is sometimes called intermediate fine-tuning. It is already known to improve performance in certain tasks.


**Strength And Weaknesses:**

# Strengths
- Improvements over BART baseline,  which has the same architecture, in some tasks.
- In some NLG cases achieves SoTA, although I've not verified every task.

# Weaknesses
- There is a major mismatch in the way the paper is written and the experimental setup. It is written to suggest that supervised pre-training can replace unsupervised pre-training, but does not have experiments to back this up. Consequently it has highly misleading claims on merits of supervised pre-training since the models are primarily pretrained in an unsupervised way.


**Summary Of The Paper:**

Considers supervised pre-training as an alternative to unsupervised pre-training, which is more common with state-of-the-art NLP models. Many (77) NLG supervised datasets are curated to create MVPCorpus which is used for supervised pre-training. However, the training setup is not designed in a way that allows us to compare unsupervised vs supervised pre-training since the backbone model is initialized with BART parameters. Thus the paper cannot answer the question which is posed, and instead shows that continued supervised pre-training using related tasks after starting with an unsupervised pre-trained model helps with downstream tasks.

The model is initialized with BART and is further 'pre-trained' using supervised/multi-task learning. The main baseline is BART of the same architecture and some variants are compared.

Fine-tuning is done in two settings: full fine-tuning, and with prefix tuning. They show that MVP does better in both settings compared to BART and achieves some SoTA results in NLG in the FT setting. Finally, it is shown that zero-shot performance is better than BART as well.


**Summary Of The Review:**

This paper should be re-written as a study of how to continue pre-training with multi-task supervised learning or properly study supervised pre-training by initializing the model from random parameters. At best, if re-written this way the paper shows how to achieve slightly better results on certain NLG tasks compared to BART with intermediate fine-tuning. The novelty is limited and the claims are incorrect.

---

> ### Author Response · Authors · 2022-11-14
> **Response to Reviewer AMUT**
>
> Thank you for your meaningful reviews!
>
> Nevertheless, we cannot agree with your viewpoints.
>
> - A major controversy is the use of pre-trained parameters rather than randomly initialized ones. To our knowledge, it is a common method for pre-training a new model using existing checkpoints. For example, UniLM is initialized with BERT, DialoGPT is initialized with GPT-2, and T0 is initialized with T5. Whether or not we use the pre-trained parameters does not affect the method that we are pre-training a model with multi-task supervised learning.
>
> - Furthermore, the major contribution of our work is the open source of the MVPCorpus, MVP models, and the improved performance on numerous tasks and datasets, just as the Reviewer 4XxP noted. Our MVP model achieves better performance than BART, with an improvement of 7.0% (in ratio).
>
> - We have described how to mix each task and dataset in Section 3.3 (the last sentence of the first paragraph), using a temperature-scaled mixing strategy following T5.

---

> > ### Author Response · Authors · 2022-12-06
> > **Response to Reviewer AMUT**
> >
> > We are still very willing to discuss the clarity of our work with you. Please feel free to contact us with any additional suggestions or questions. We will keep doing our best to provide you with answers.

---

### Official Review · Reviewer_VRii · 2022-10-25

**Confidence:** 4
**Correctness:** 2
**Technical Novelty And Significance:** 1
**Empirical Novelty And Significance:** 1
**Recommendation:** 3

**Clarity, Quality, Novelty And Reproducibility:**

- There is some part quite hard to follow. How do you train  (MVP+R) model?
- the originality is marginal


**Strength And Weaknesses:**

### Strengths
- Collect a significant amount of training data for seq2seq tasks.
- Reconfirm empirically that using more data is always useful

### Weaknesses
- The contribution of the paper is too thin in my opinion (i.e., showing that using more data is useful)

- On *Generaliability* study: I don’t really understand this setup. The authors call it unseen tasks but then the  model is fine-tuned using the annotated data of the task. How is this different from taking any pretrained model and finetuning it on the same data? My understanding here is that the “unseen” part means that the tasks and data weren’t in the continued train step. But calling that _generaliability_ seems to be an overstatement to me. Is that what we all do with pretrained models?
- There are no human evaluation results. Evaluating language generation is tricky, using automatic metric alone is not sufficient.
While measuring by automatic metric, it shows that supervised pre-training with  more data is useful. There are many datasets with different characteristics. Does it hurts the model in a way that we don’t know yet? What type of hallucination in language generation when the model is trained on diverse datasets? Does the nature of informal text in the Personal Chat dataset have negative influences on formal text CNN/Daily Mail. I think there are many important questions about mixing random datasets in the context of language generation that have not been answered. The small improvement in automatic metrics do not fully justify the approach.



**Summary Of The Paper:**

The paper propose to continue pre-train a pre-trained seq2seq model using annotated data for generation tasks. The authors collected 77datasets over 11 diverse NLG tasks for the continued train step. Evaluation shows good results.


**Summary Of The Review:**

In summary, the contribution of the paper is thin. The evaluation is not convincing (lack human evaluation). There are potential issues of concatenating multiple language generation datasets in the supervised pretraining step.

---

> ### Author Response · Authors · 2022-11-14
> **Response to Reviewer VRii**
>
> Thank you for your meaningful reviews!
>
> - The major contribution of our work is the open source of the MVPCorpus, MVP models, and the improved performance on numerous tasks and datasets, just as the Reviewer 4XxP noted. Our MVP model achieves better performance than BART, with an improvement of 7.0% (in ratio). We believe that the datasets, models, and codes will promote NLG-related research.
>
> - Regarding generalizability, thank you for pointing out this issue. We will change the section title to " Fine-Tune on New Tasks".
>
> - For the human evaluation, we believe it is beneficial, and we will add it in a future version. And your concerns about automatic metrics are also meaningful. However, we have conducted extensive experiments (35 datasets in total), hence we think the improvement can be representative. It is also difficult and expensive to conduct human evaluation on so many datasets.
>
> - As described in Section 4.1, the MVP+R model is the MVP model with randomly initialized prompts.

---

> > ### Author Response · Authors · 2022-12-06
> > **Response to Reviewer VRii**
> >
> > We have added the human evaluation results, which prove our MVP's effectiveness. We are still very willing to discuss this with you in order to fully address your concerns. Please feel free to contact us with any additional suggestions or questions. We will keep doing our best to provide you with answers.

---

### Author Response · Authors · 2022-12-06
**Results of Human Evaluation**

We conduct a human evaluation on four tasks, including text summarization (CNN/Daily Mail), data-to-text generation (WebNLG), open-ended dialog system (PersonaChat), and story generation (ROCStories). Following the practices of van der Lee et al. (2021) [1], we utilize a stratified sample of 100 inputs of low, medium, and high frequency for each task. We invite six human judges to evaluate the generated texts of the inputs by MVP and BART. For each instance, they can see one piece of input text and two pieces of generated text without knowing which model it is from. Then they need to choose which one is better or choose a tie according to the text's fluency, informativeness, consistency, and task features.

The following table showcases the proportions of "MVP wins," "Ties," and "BART wins" for each dataset. $\alpha$ denotes Krippendorff's $\alpha$, which measures the inter-annotator correlation of human judges. From the results, we can see MVP can generate overall better texts than BART on these four datasets.


| Dataset        | MVP wins (%) | Ties (%) | BART wins (%) | $\alpha$ |
| -------        | --------     | ----     | ---------     | -------- |
| PersonaChat    | 35.33        | 34.00    | 30.67         | 49.70    |
| ROCStories     | 46.50        | 11.33    | 42.17         | 27.40    |
| CNN/Daily Mail | 46.50        | 10.67    | 42.83         | 23.83    |
| WebNLG         | 32.17        | 45.67    | 22.17         | 50.93    |

---

### Decision · Program_Chairs · 2023-01-20

**Decision:**

Reject

**Justification For Why Not Higher Score:**

Reviewers are not supportive of the paper. Contribution is also quite weak here.

**Justification For Why Not Lower Score:**

There isn't really anything lower than a reject.

**Metareview: Summary, Strengths And Weaknesses:**

Reviewers were not supportive of the paper and scores are pretty low.

I also skimmed the paper myself. Many of the usefulness of multi-task pretraining or finetuning have already been shown in ExT5/T0/Flan so I was also not able to find anything interesting in this paper.